# Public perceptions on carbon removal from focus groups in 22 countries

Sean Low [1] ✉, Livia Fritz [1], Chad M. Baum [1] & Benjamin K. Sovacool[1,2,3]

Carbon removal is emerging as a pillar of governmental and industry commitments toward achieving Net Zero targets. Drawing from 44 focus groups in 22 countries, we map technical and societal issues that a representative sample of publics raise on five major types of carbon removal (forests, soils, direct air capture, enhanced weathering, and bioenergy with carbon capture and storage), and how these translate to preferences for governance actors, mechanisms, and rationales. We assess gaps and overlaps between a global range of public perceptions and how carbon removal is currently emerging in assessment, innovation, and decision-making. In conclusion, we outline key societal expectations for informing assessment and policy: prioritize public engagement as more than acceptance research; scrutiny and regulation of industry beyond incentivizing innovation; systemic coordination across sectors, levels, and borders; and prioritize underlying causes of climate change and interrelated governance issues.

Carbon removal – the development, upscaling or utilization of a diverse range of carbon sinks – is emerging as a pillar of governmental and industry commitments toward achieving Net Zero emissions reductions targets[1,2]. Approaches range from interventions in agriculture, forestry, and ecosystems management to large-scale engineering systems, spread across terrestrial and marine environments as well as urban and rural communities. Carbon removal at scales projected for Net Zero targets would implicate polities, geographies, and sectors across the global North and South[3,4].

Public engagement – particularly through practices of deliberation, inclusion, and reflectiveness – is essential to gauging the feasibility and governability of heterogenous and often immature carbon removal options. Such practices have been diversely deployed in climate change[5], technology and environmental assessment and governance[6,7], energy and climate policy[8]. These literatures have all demonstrated capacities for mapping 'situated' perspectives (bottom-up actor- and locale-specific; in distinction to top-down, systemic, global-planning), anticipating the 'fit' between emerging issues and local context, and developing societal capacity for further inquiry and learning-by-doing. Assessments of carbon removal also cross socio-political, technological, and ecological

boundaries, and require existing and novel sectors and practices to be integrated – comparable examples in climate action include large-scale renewables[9], shale gas[10], and reducing emissions from deforestation and forest degradation in developing countries (REDD+)[11]. Following these studies, we deploy deliberative engagements to anticipate key challenges and provide input for shaping legitimate governance processes.

To ground our study, we engage with the carbon removal public perceptions literature[12] to see how a broadened set of publics nuance its most prevalent findings. We draw upon three key areas. The first describes rationales underpinning preferences for and against particular approaches, including: the naturalism bias favouring biogenic approaches[13,14] and linked aversion to 'tampering with nature'[15,16]; concerns about land-use trade-offs for forestry and agricultural management practices, including the bioenergy component of bioenergy carbon capture and storage (BECCS)[17–19]; concerns about storage locations and leakage for direct air capture and carbon storage (DACCS)[20,21]; the grey area between rejection based on pollutive siting versus Not-In-My-Backyard-ism[22]; and comparisons to analogical or related technologies such as shale gas[23] or carbon capture and storage (CCS)[24]. Secondly, studies point out public capacities to

[1]Department of Business Development and Technology, Aarhus University, Birk Centerpark 15, 7400 Herning, Denmark. [2]Science Policy Research Unit (SPRU), University of Sussex Business School, Jubilee Building, Arts Rd, Falmer, Brighton BN1 9SL, UK. [3]Department of Earth and Environment, Boston University, 685 Commonwealth Ave, Boston, MA 02215, USA. ✉e-mail: sean.low@btech.au.dk

assess synergies or trade-offs of upscaling carbon sinks with wider climate and sustainability action[20,21,25,26], including concern over the development of excuses to further delay comprehensive emissions reductions[27]; and conceptions of justice and equity that drive preferences[28]. Finally, studies also delve into policy and governance, and demands and conditions under which further research and field experiments[20], supply chain development[19] or different modes of policy[29,30] become relevant.

In this work, we map prospective benefits and risks and corresponding governance regarding five major types of carbon removal, engaging with 44 focus groups (1 urban, 1 rural) in 22 countries worldwide – in Europe (Austria, Germany, Italy, Norway, Poland, Sweden, Switzerland, Spain, United Kingdom), North America (United States), Latin and South America (Brazil, Chile, Dominican Republic), Africa (Kenya, Nigeria, South Africa), the Middle East (Saudi Arabia, Turkey), and the Indo- and Asia-Pacific (Australia, China, India, Indonesia). Our engagements cover five major types of carbon removal: (a) afforestation and reforestation as an entry to diverse marine and terrestrial ecosystems management practices; (b) soil carbon sequestration (including but not interchangeable with biochar); (c) DACCS; (d) enhanced weathering (EW); and (e) BECCS. We map key technical and societal issues that focus groups raise on particular approaches, and how these translate to preferences – and varying degrees of trust – for named actors (expert networks, civic organizations, countries, and intergovernmental frameworks), mechanisms (kinds of assessment, funding, and policy), and rationales (underpinning intents for governance). Our discussion further maps gaps and overlaps between how publics view the prospective upscaling of carbon removal approaches, and how they are emerging in global assessment, innovation, and decision-making. In conclusion, we outline key societal expectations for informing assessment and policy. Our intent is to derive a global, 'horizontal' benchmark of hopes, concerns, and expectations of assessment and governance for further deliberation.

## Results

We structure our results in three portions. First, we highlight hopes, concerns, and governance relevant to each carbon removal approach (afforestation and reforestation, soil carbon sequestration, DACCS, enhanced weathering, and BECCS). Second, we highlight potential synergies or trade-offs with wider climate and sustainability action. Finally, we move to governance dimensions – roles, processes, and rationales of assessment, industry and innovation, publics, and government – that cut across carbon removal approaches.

Our reporting compromises between two needs: summarization versus allowing participants to speak in their own words, thereby demonstrating greater nuance, ambivalence, and discursive interaction. In text, we deploy summary descriptions of themes, but make use of extensive quotations in Supplementary Table 1, Supplementary Table 2, and Supplementary Table 3 to give a sense of the diversity and depth of deliberations. Table 1, Table 2, and Table 3, included in text, are shorter versions containing a selection of the most illustrative themes and quotations.

### Afforestation and reforestation

All focus groups supported approaches for making resilient or (re) expanding ecosystems as carbon stocks, often comparing them favorably to the resource and infrastructural costs of chemical or engineered systems (especially DACCS, and occasionally EW). In deliberation, terrestrial afforestation and deforestation were often expanded to marine environments (e.g. mangroves), as well as to ecosystems management more broadly. However, the strongest strands of conversation remained on forestry.

Participants were strongly swayed by preferences for ecosystems-based approaches, which were often described as being "natural", or a "part of nature". Sometimes, participants attached this preference to the inchoate but resonant lens of reversing industrialization or the human footprint. But a preference for biogenic approaches was also concretely coupled to familiarity – accelerating known practices,

## Table 1 | Carbon removal approaches

| | |
|---|---|
| Afforestation and reforestation | Combat deforestation (Kenya, Rural): <br> I think we need to campaign for people that are cutting down trees and we need to campaign for people regrowing more trees. I think our government should look at more opportunities for creating things like jobs... This is because people tend to cut down trees in search of better living, so they can sell it to the market. <br> Spatial trade-offs with food crops (Nigeria, Rural): <br> ... if we need so much space – because the amount of CO2 we are talking about is not a small one – so that means we need land to plant trees to absorb the CO2, just like where we talk about the land to plant bioenergy crops... with time we will run out of food. There would be food shortages because we would need land for the vegetation. |
| Soil carbon sequestration | Enhance agricultural capacity (Dominican Republic, Rural): <br> Well, that has an impact on the economy because there is more food production and I also believe that a country with food production has less scarcity, so it helps in every sense of the word. |
| Direct air capture and carbon storage | Benefit for first mover industry and innovation (Switzerland, Urban): <br> Finally, the manufacturers of the filter systems... they can make a big business out of it... with the storage facilities, would certainly also be interesting for business and industry, because ultimately it will pay off again if these products are promising and if they can be produced and sold. <br> Siting, leakage, and pollution exports (Austria, Rural): <br> And of course, the location of the storage is problematic... You're not doing something directly for your country, so no one wants to take on the drawbacks because everyone will benefit from the advantages... We're talking about huge installations on these big areas, so it has to benefit the people who live there – you can't just build it. |
| Enhanced weathering | As an extension of mining (South Africa, Rural): <br> Most probably we should be cleaning up those quarry stuff.., I think we should recycle available wastes or rocks and play with that. But as for going to blow up completely new rocks? I'm not for that. <br> As soil enhancement (Nigeria, Urban): <br> Actually in local areas we don't believe in chemical manure, we believe in natural manure... I should know how it works first. I think samples should be carried out, they should identify the type of rock that would be needed. |
| Bioenergy carbon capture and storage | Less carbon intensive fuel, Energy security (Australia, Rural): <br> I think transport, like the bioenergy, if that was a fuel source for cars and industry and an ultimate one, I think that it would make us more secure as well as self-sufficient. <br> Spatial trade-offs with food crops (India, Urban): <br> Because the farmers would be thinking that growing these kinds of plants is beneficial, we shift from food crops to CO2 plants, so that the food industry will also be affected in the future. So, we would face some food scarcity also. |

**Table 2 | Climate and sustainability action**

| | |
|---|---|
| Underlying causes of climate change and unsustainability | China, Urban:<br>We need solutions which address the underlying/fundamental problem. The environmental problem was caused by this generation – excessive use of energy, rapid industrialization, uses of gasoline cars, air conditioning…all things which have affected the environment negatively. If the underlying problem is not addressed, the negative impacts will be passed to our sons and grandsons (future generations). We need to be committed to solve the underlying problem. |
| Mitigation deterrence | Dominican Republic, Urban:<br>Well, it would take a problem out of their minds, also for businessmen who still won't want to change from fossil fuels to something more eco-friendly. For example, we are trying to change everything to solar panels, electric cars – but since it's not that beneficial for corporations, there are some restrictions. |

**Table 3 | Cross-cutting governance dimensions**

| | |
|---|---|
| Assessment | Triple helix (Indonesia, Urban):<br>Maybe these scientists, the people from educational institutions, they'd conduct research to know whether it could go on/run or not… Secondly, it's the government. Maybe they're more about the policy… Also, the socialization (information and consultation) of those things to the people. |
| Publics | Local consultations (Germany, Rural):<br>I always think it is a matter of whether you include the entire population or the affected population. We talked before about the agri-culturalists, who of course have to be involved because it affects them directly. With direct air capture… if this big plant is placed in my neighbourhood, I would like to be involved in the decision making. I do think that the affected populations should be included, just to make it functional long-term.<br>Not passing the burden to citizens (Sweden, Rural):<br>… governments can take responsibility on such; it is not maybe every single individual that has to take responsibility. To look at their carbon dioxide budget… there are big institutions who do that. Countries that collect it and process it for every one of us… That it can be controlled in larger scale. |
| Industry and Innovation | Control industry (Brazil, Urban):<br>We know that companies, as a reflection of people, don't do many things voluntarily. I believe a law should pass to oblige companies to care more about the environment and storage and capture carbon dioxide, even using financial retributions (sic). I think the first step is passing a law. We don't see companies voluntarily solving this problem, only a few do.<br>Facilitate industry, corporate social responsibility (India, Urban):<br>(On DACCS) I would like … industry forums to be done, lot of intra activity among participants, maybe a group newly created where in you have these heads of various ministries and various public sector undertakings and various bodies such as the Confederation of Indian Industry; Tata, Mitra and all the big names in the industries… And of course, you will require the government also to be a part. |
| Government | Systemic coordination (Poland, Urban):<br>In some regions of the world restoring and cultivating vegetation would probably work better. In others – storing carbon… DACCS and BECCS could not be used everywhere, because it would have to be in former industrial regions, where there are mines, or some oil wells that are out of commission… It should be a global rather than local effort, adapting to the local possibilities of those places where these projects could be implemented.<br>Trust for, and need to support local governance (Italy, Urban):<br>I believe that regions as well should have a role. They can even implement supporting policies as it happens. For example in Trentino Alto Adige, people receive incentives to change into green activities. They can distribute incentives taken from the European Community, through channels such as the EU. In order to do things we need money. Besides all the nice words, we know that municipalities do not have money. We should create policies that support such activities. |

initiatives, and regulations/incentives. Participants further associated naturalism and known practice with agency: these approaches were easy to technically grasp, had varied and distributed applications within and beyond their own countries, and ordinary citizens could personally participate in upscaling (e.g. as farmers or landowners), through civic initiatives, or alongside municipal or local government. Some highlighted co-benefits for biodiversity (rewilding, national parks), socio-environmental resilience (soil erosion, wind-breaks), aesthetic and religious value, and for food provision.

Proposed governance focused on opportunities to harness the diversity of application. Groups – especially in tropical forested countries – connected afforestation and reforestation efforts to banning or disincentivizing deforestation, establishing protected zones for key or iconic ecosystems (the Amazon, or wetlands), or restoring deforested areas and re-purposing brownfields. Avoided deforestation is not technically classified as a form of carbon removal, but focus groups often conflated this with afforestation and reforestation as part of general forestry management efforts. Strong pluralities across North and South highlighted the need for diverse land-use regulations coupled with private-public initiatives (e.g. monitoring and halting illegal logging; zoning / land-reallocation for green areas; tree-planting campaigns; strengthening small-holder ownership), direct state intervention (buying brownfields; compensating for lost income; supplying seeds), and especially, economic incentives and consultation campaigns for landowners to prioritize carbon stocks. Urban opportunities were noted across North and South contexts: rooftop and vertical gardens, city zoning and construction laws requiring green areas and components, and co-benefits for urban health and air pollution. Much questioning surrounded the scale and sequestration potential of urban carbon stocks.

Regarding concerns: strong pluralities across the global North and South highlighted two spatial trade-offs in using land for carbon stocks. The first was a trade-off with food crops. Fearing food insecurity, groups referenced the bioenergy-driven food versus fuel trade-off; or the need for income stability, given a lack of financial incentives to choose against growing cash crops. The second trade-off referenced competition with residential development, driven by widespread perceptions of population growth, with (often, global South) participants highlighting insufficient housing developments, and others (often, global North) noting distributed, space-intensive private property. Both trade-offs motivated concerns of property seizure in a smaller number of groups, ranging from corporate versus smallholder land-grabs familiar in bioeconomy conflicts to 'eminent domain' appropriations. The impermanence of carbon forestry driven by profit motives emerged as a concern from a small plurality across all regions, though most strongly represented in European groups: in the absence

of proper monitoring, lumber companies might quickly deforest replanted areas, exacerbated by illegal logging and governmental corruption in granting permits.

The protection of the Amazon rainforest was seen as a keystone example of all these issues: an iconic ecosystem's management as carbon stocks, socio-economic co-benefits, and deforestation under threat of corrupt governance and agricultural and ranching pressures. Interestingly, these issues were raised amongst European groups; Brazilian groups themselves emphasized the Amazon less while criticizing deforestation in the global North.

## Soil carbon sequestration

Just as afforestation and reforestation were seen as proxy for ecosystems management approaches, soil carbon sequestration was synonymous with agriculture management efforts. The same inclinations towards ecosystems-based or so-called "natural" approaches, accelerating known practices, and agency – unsurprisingly, for farming communities and agribusinesses – were present. A very small plurality noted initiatives for alternatives to importing artificial fertilizer such as organic fertilizers and composting, waste biomass-gathering, and permaculture.

Groups from a majority of countries – particularly from the global South – saw governance efforts as a matter of accelerating known practices through education and consultation, and significantly, economic incentivization. Deliberations highlighted the need for governance to cater to the diversity of agricultural actors and contexts across the global North and South: ranging from subsistence smallholders to family farms to corporate farms; or various parts of agribusiness supply chains, from seeds and fertilizer supply, to production, to educational and training facilities.

A priority from a vast majority of countries was on the capacity of soil carbon sequestration to preserve or enhance agricultural capacity, citing food security issues. Concerns over spatial trade-offs with growing carbon stocks (e.g. see perspectives on using land for afforestation / reforestation) could also be filtered through this lens, though a small number of groups connected forestry and agricultural management as part of the same land-use management initiatives.

## Direct air capture and carbon storage

Compared to biogenic approaches, participants found DACCS more difficult to grasp technically, while seeing its infrastructure, energy needs, and siting as emerging from centralized, supply-driven cooperation between government and industry. A sense of agency and direct participation was muted.

Participants from a vast majority of countries envisioned that benefits would accrue to first-mover innovators and manufacturers of DACCS systems and carbon capture storage and utilization (CCUS) applications, as well as the creation of new jobs – some stated this as a hope, and others as an inevitable development of government-industry ties. Focus groups from countries with more positive perceptions of government-industry collaborations (e.g. visibly successful state-owned enterprises, and/or a highly functioning, well-regulated technology sector) tended to speak of this more favourably. These included China, India, Saudi Arabia, Norway, and Switzerland. Conversely, participants with more negative perceptions of government-industry collaborations worried about the social costs of developing a high cost, high energy infrastructure. These participants were from a North-South crosscutting plurality, regarding corruption or disproportionate government support for extractive industry. Another concern was on unequal technological and financial capacities, where state funding for DACCS innovation would compete with other uses of public funds – brought up equally in global North groups as in global South. There was discussion of technology transfer: groups from a small number of global North countries argued that they should lead DACCS development and

distribution – matching a demand made by a larger number of global South groups.

More so than global South groups, global North groups grappled with whether concern over DACCS infrastructure siting constituted NIMBYism. European participants criticized what they saw as NIMBYism in their own countries, most commonly with reflexive reference to the antecedent of siting wind turbines. However, concerns were more commonly couched in terms of what researchers have termed hazardous siting: noise and physical pollution from infrastructure, and above all, physical leakage of carbon from storage or transportation. This was held by groups in an overwhelming majority of global North countries, with pluralities in global South countries.

The grasp of technicalities varied: focus groups questioned the safety of storage in deep sea or subterranean reservoirs, or in temporary facilities and transportation. Focus groups from countries with extensive oil and gas infrastructure (US, Nigeria) highlighted the possibility of sabotage due to theft of gas or even warfare. Groups from half the global North countries recognized what researchers have termed pollution exports to potentially low-regulation jurisdictions, particularly in the global South, as an important component of hazardous siting.

Participants cited the need to connect debates on technology transfer and moving pollution abroad, ensuring that safety standards accompany carbon waste movement. Nuclear waste, biohazards, and plastic waste were common analogies through which leakage and pollution exports concerns were filtered. Interestingly, the pollution export concern was raised in almost no groups from global South countries (though leakage concerns were).

Groups from a majority of North countries and a strong plurality from emerging South countries cited the energy costs of DACCS as a concern, with a key dimension being energy efficiency. Participants questioned the trade-off in reducing emissions with the same power sources that create emissions, highlighting parallel systems (e.g. electric vehicles) in which fossil energy could be disguised and prolonged within a wider narrative of reducing emissions. As a result, what researchers have labelled as the polluter pays principle stood out as a governance rationale – held by groups from half the countries across both North and emerging economies. In this view, polluting industries – e.g. the oil and gas industry – should pay for DACCS innovation, connecting to fears of energy and financial costs being passed down to citizens, and a sense of the culpability of high-emissions industries and facilitative government-industry complexes.

## Enhanced weathering

With this set of approaches, most groups found the technicalities of carbon drawdown and storage difficult to grasp. All themes regarding this approach emerged from small pluralities. Some situated EW between biogenic and engineered carbon removal: more natural (accelerating a 'natural' process) or reliant on existing industry and infrastructure (farming or mining) than DACCS, but at the same time having unfamiliar energy and siting demands. Hopes and concerns depended on whether particular focus groups envisioned EW primarily as an expansion or refinement of mining operations, or for additional benefits through agricultural management. Ocean alkalinization was not a focus of this study.

Seen through the lens of expanding mining operations – the theme with the largest plurality, though again with a majority in the North – participants questioned the locations, scale, and energy costs of mining operations as well as transportation to sites of deployment. Social and ecological siting impacts were also raised: whether new, numerous, and expansive mining locations would be warranted and what geographic formations and ecosystems would be affected, as well as the uncertainties of what EW would represent on-site. Many questioned spatial trade-offs in the use of coastal and terrestrial natural areas in both mining expansions and EW sites – for farming,

(traditional, indigenous) ecosystems services, and recreation. Some highlighted the role of the mining industry and the plausibility of land-grabs, hazardous siting, and disproportionate change inflicted on local communities mirrored in discussion of other carbon removal approaches, citing the need for consultation and compensation. Participants from countries with large mining sectors (e.g. Australia, South Africa), spoke to purposing EW for recycling mining waste and cleaning up existing quarries.

A small but North-South cross-cutting number of groups saw co-benefits for EW as soil enhancement, calling for further assessment of the types, locations/sources, and efficacy (compared to other kinds of fertilizer) of rock needed. In some cases, participants made sense of EW by conflating it with soil carbon sequestration.

## Bioenergy with carbon capture and storage

In comparison with DACCS, there was a more grounded familiarity with bioenergy as a system component from participants in countries with bioenergy and biofuel sectors, who cited synergies with and expansion of known practices and regulations/incentives. The capacities of bioenergy as a less carbon intensive fuel source for transportation, residential areas, and industries were debated – as well as in terms of energy security, as a replacement or supplement for (imported) hydrocarbons. These deliberations took place mostly in US and Europe groups, with one Brazil group the lone entry from global South countries.

However, familiarity with bioenergy also led to discussion of how BECCS could mirror known food versus fuel conflicts. These were prominent in Dominican Republic, Kenya and Nigeria. Participants from rural backgrounds leveraged deliberations on afforestation/reforestation and soil sequestration approaches to discuss the use of farmlands, the types of bioenergy crops demanded, and the role of GMO crops. A couple of groups grappled with the possibility of using waste crops and abandoned land, but this was not prevalent.

A small number of groups (with only one from the global South) raised hazardous siting concerns, questioning the pollution caused by biofuel plants and location amongst marginalized communities, or a lack of public consultation or information. As with EW, there was an unclear technical grasp of storage, with some filtering concerns about leakage through similar concerns about DACCS, and others to soil toxicity concerns regarding soil carbon sequestration. The polluter pays principle emerged in a single group discussion – far less commonly than for DACCS.

## Climate and sustainability action

Caution on carbon removal approaches was most prevalently tied to localized hazardous siting issues and trade-offs, then branching out into imposed costs or needed incentives at the local-to-national level, and further still to food security or pollution exports at the multi-national level. However, a small number of groups pointed out that carbon removal does not address what they see as the underlying causes of climate change and unsustainability: industrialization and economic development imperatives, the carbon economy, unsustainable resource usage, and profit-seeking motives. Seeking co-benefits, many groups deliberated on how carbon removal might be integrated into the governance of local-to-national concerns, with implications for climate and sustainability governance. Aforementioned examples include deforestation and agricultural productivity/food security (biogenic approaches and the bioenergy component of BECCS), or both leveraging and controlling extractive industries (oil and gas for DACCS; mining and agriculture for EW), or local socio-environmental conditions, from health and pollution to climate impact vulnerability (all approaches).

A few groups across North and South acknowledged that carbon removal might buy time for more comprehensive decarbonization efforts, with a smaller subset questioning whether engineered approaches like DACCS might further buy time for the regeneration of natural carbon stocks. On the other hand, a larger number of groups mostly in the global North extended deliberations on the need for addressing underlying causes of climate change to note what experts have termed mitigation deterrence:[31] that carbon removal might create concrete disincentives towards decarbonization, with personal (e.g. consumer choice, individual footprints) as well as commercial and industrial motives given. Some warned against taking a technical, technocratic approach to systemic societal problems or the unexpected or additional consequences of building global carbon removal systems –tackling climate and sustainability issues with the same logic that created them.

## Assessment

Participants in every focus group referenced a high degree of trust in expert-driven assessment, with a lesser but significant number of groups referring to a complex linking inter-disciplinary assessment with governmental decision making and industry and business actors (what experts term the 'triple helix' model[32]). Groups in a majority of countries highlighted the role of national scientific bodies or specialized government departments and ministries to bridge sectors and launch assessments. Conversely, strong pluralities across North and emerging economies attached the demand for expert-led assessment to a mistrust of leading governmental or industry involvement in foundational studies of feasibility, citing antecedent complexes in extractive industries and greenwashing.

Participants also referenced the need for technical expertise tailored to specific approaches. According to groups in a majority of global South countries, and a smaller number of global North groups, field tests should assess technical feasibility and safety/impacts, mostly regarding DACCS and BECCS. At the same time, participants – with roughly the same distribution – highlighted a strong demand for strengthening public and stakeholder engagement, to foster education and mutual learning (all approaches), innovation (e.g. for DACCS), and learning-by-doing (e.g. for more distributed biogenic approaches). Accordingly, universities and educational institutes were seen as potential innovation hubs as part of the triple helix model, as well as spaces for knowledge dissemination and training.

## Industry and innovation

Groups commonly perceived a first-mover advantage for (national) industries with capacity to take advantage: e.g. energy provision, innovation, and manufacturing for DACCS, BECCS, and EW (the mining component), which are viewed more through centralized, supply-driven pathways of scaling and governance. Groups generally recognized the diversity of logistical and political roles and actors. Envisioned roles are as diverse as organizing labour and supply chains for biogenic approaches (e.g. tree-planting campaigns), to 'triple helix' or hybrid collaborations with governments, universities, media, and publics for information campaigns, demonstration projects, or to construct governance. Envisioned actors include innovation-driving start-ups to state enterprises representing major manufacturers or extractive (fossil fuels, mining) or agro-industries (bioenergy, agriculture).

At the same time, many participants placed culpability and responsibility for a leading contribution to upscaling carbon removal on industries, due to mistrust of commercial motives in driving deforestation and impermanence in carbon forestry, as well as emissions from extractive industries, the convenient fit for bioenergy and fossil fuels industries to benefit from BECCS and DACCS innovation, and 'Polluter Pays' as a rationale for industry financing of high cost, high energy options.

Participants made a virtue of necessity in spurring industry and innovation through incentives and regulations, in two variants. A North-South crosscutting plurality sought to control industry through

public and governmental oversight. The second aimed to facilitate industry, exhibiting greater trust in government-industry collaborations and state-owned enterprises. In the latter view – held more prevalently in groups from Northern and emerging economies – industries rich in material, manpower, and intellectual resources should voluntarily lead, fostering what experts – and even some public participants – term corporate social responsibility. Nor was this solely related to DACCS innovation: some groups saw wide-reaching roles for marshalling logistics and labour for tree-planting or land provision.

### Publics

Public action was most often underpinned by agency and willingness for citizens to engage to the fullest extent possible. Another significant rationale was that those affected should have a voice in decision-making, connected to participation in distributed upscaling of biogenic approaches, as well as to consent over hazardous siting of engineered systems.

Publics saw for themselves three broad roles. The first was a diverse range of proactive self- or community engagements: in learning and knowledge dissemination (through social media, or local/community information campaigns); participating directly in hybrid, distributed and decentralized initiatives for urban and rural ecosystems restoration (e.g. tree-planting), taking up soil carbon approaches in farming communities; consumer choice (gold and green standards for products, generating offsets); funding initiatives through donations, or more contestably with regard to high-cost/energy options, taxes. The second was the most commonly cited, though more passive: relying on government and industry to initiate information campaigns or educational programs through the national media and state / education institutions. Both roles were voiced by groups from all or a vast majority of countries.

The last avenue, held by a lesser but globally cross-cutting majority, recognized the need for governments and industries to consult local stakeholders on design and siting issues. This bridged passive and proactive elements, and was conditioned by approach. There was a strong sense of agency regarding biogenic approaches, with local (farming, rural, landowning) communities able to offer inputs for co-design of carbon stocks management and spatial trade-offs. For DACCS (and to a lesser degree, EW and BECCS), publics were more concerned with consent or compensation regarding pollutive infrastructures.

Lesser pluralities compared various carbon removal approaches against personal efforts to reduce their own carbon footprint. These included recycling, reducing plastic or meat consumption, using low-carbon transportation, and various other consumer choices. This frame of reference might be recognized as a means by which citizens feel empowered to act in the face of a systemic problem such as climate change, or carbon removal as climate action. Some groups from the global South cited such individual efforts as demonstrative of broader climate and environmental priorities, questioning if public engagement and policies on carbon removal might have stronger traction in (certain) global North countries as compared to their own. A couple of European groups used this logic to deliberate on whether biogenic approaches are preferred to engineered approaches, given more familiar, diverse, and distributed dimensions, with clearer points of entry to public engagement or consultation, and connections to local laws, bodies, and initiatives.

At the same time, a stronger plurality across North and emerging South countries argued that 'individualization' engenders the illusion that issues demanding 'supply-side' policy action can be tackled by 'demand-side' consumer choice. Most groups used this to refer to climate action more broadly. But in relation to carbon removal, some were wary that leveraging individual responsibility should not shift the burden of action away from industrial or corporate polluters (the Polluter Pays principle) or governments.

### Government

Strong pluralities cutting across the global North and South prescribed the key rationale for government(s) as providing coordination, between sectors (land-use, extractive industry; assessment processes and publics; levels (between ministries, agencies, research institutes, municipalities); regions (between sub-national jurisdictions, geographies); and at the multilateral level (regional or global coordination).

Some expressed skepticism about effective multilateralism, but groups from a strong majority of North and South countries highlighted systemic coordination (the authors' term) needed to manage the procedural and distributive politics of funding, technology development, siting, infrastructure maintenance, and pollution exports. The motivation behind governmental coordination was couched around national or regional advantages regarding different carbon removal approaches, given resource- and geography-specific criteria, coupled to the need for an integrated multilateral portfolio of carbon removal approaches. These included: the presence of bioeconomy, forestry, and agrarian sectors for biogenic approaches and the bioenergy component of BECCS; technology and financial capacity and/or oil and gas industry and geological reservoirs for DACCS; the mining sector and appropriate sources of materials for EW. A smaller plurality in global North countries acknowledged the need to take account for historic responsibility for emissions and deforestation (reflecting expert and policy conversations about Common but Differentiated Responsibilities) – but this was not prevalent in global South groups.

Unsurprisingly, some European focus groups mentioned the European Union as a key regional coordinating body with clear capacities and mandates, though trust in its institutions could vary. Groups across majorities of North and global South countries named the United Nations system – most commonly, the UN Framework Convention on Climate Change (UNFCCC). Participants sometimes cited bodies with which they were familiar with from antecedent trans-boundary governance issues – such as the World Health Organization, regarding the Covid-19 pandemic – as templates for expedited multilateral assessment and coordination.

Perspectives underpinning trust in other levels of government and governance varied. Some voiced low trust in national government on diverse issues. Some groups – perhaps, from countries with strong state capacity and/or communitarian structures (e.g. China, Saudi Arabia, Norway) – expressed fuller confidence in the national government as the natural locus for societally-mandated action, and particularly for the control or facilitation of innovation and industry. The search for crosscutting rationales was unedifying. The theme most clearly related to (engineered) carbon removal was on trust in industry-government collaboration. Otherwise, rationales were specific to national context, referencing issues as diverse as corruption; (mis)management of longstanding issues (e.g. housing or land-use planning) or recent crises (e.g. Covid-19; Brexit); comparisons of governance against that of other countries; contestation over the performance or jurisdictions of different levels of governance (e.g. national versus EU-level; federal versus provincial level); and most inchoately, the culture of national politics.

Meanwhile, groups ubiquitously expressed greater trust in local governance capacities to understand differentiated siting concerns (all approaches), capacities for learning and upscaling (particularly biogenic approaches), and to organize assessments and consultation. Examples ranged from formal municipal governments and village councils to more traditional (often rural) forms of representation (e.g. cultural/religious leadership), to ad-hoc or sustained forms of public engagement.

## Discussion

Focus groups displayed a preference for biogenic carbon removal over engineered approaches. In doing so, they confirmed the well-documented inclination towards naturalism[13,16] and familiarity[18,33]. The

dangers of relying on framings of naturalism are well parsed[34], and we focus below on other challenges.

Groups in all global regions highlighted building support for biogenic carbon stocks through locally tailored developmental co-benefits and consultation/demonstration campaigns. Supportive policy is emerging and documented most strongly in the EU and US, repurposing the agriculture and land-use sectors for carbon sequestration[35,36]. EU and US frameworks broadly dovetail with focus groups preferences for economic incentivization, clear governmental strategy to facilitate diverse local operationalization, systemic coordination (in the EU), and to the degree that groups were concerned about impermanence of biogenic carbon stocks, (credible) certification schemes. There is a policy need to face challenges voiced by groups in tropical forested countries (e.g. Brazil, Indonesia, Dominican Republic) or facing deforestation pressures (Nigeria), or countries with large agrarian sectors.

However, expectations regarding this converging preference for biogenic approaches should be tempered. Long-term National Climate Strategies submitted to the UNFCCC largely rely on biogenic approaches to compensate for emissions envisioned as residual or hard-to-abate[37]. This creates the potential for the land-use sector to speculatively cover for industrial emissions[38] in an unfolding policy context where 'hard-to-abate' is a matter of argumentation and lobbying[39], further generating incentives to delay decarbonization[40]. Limited removal potential and vulnerability to re-emission – e.g. due to losses from global warming processes or agriculture and urbanization – are significant downsides[41].

Moreover, there is a latent international dimension of inequity and burden-shifting, with the greatest (modelled) capacity for biogenic sequestration in tropical forested countries[42,43]. REDD+, the financing mechanism for projects in the global South to reduce deforestation[44], as well as carbon forestry and voluntary carbon markets, have chequered histories in emissions accounting[45]. Indeed, regarding carbon markets: offsets were sparingly mentioned in European groups and not at all in global South groups, hinting at a profound gap between policy and public awareness. Negotiations to develop rules for international carbon credits and potentially incorporate REDD+ are ongoing over Article 6.4 of the Paris Agreement, as well as for bilateral trading of credits in Article 6.2.

A related concern held by a plurality of focus groups is on food security. This is well reflected in the public perceptions literature regarding the trade-off with food crops in forestry approaches and BECCS' bioenergy component[13,17–19,22,29,30,46], or conversely, soil-based approaches' potential to enhance agricultural capacity[33,46–48]. The nexus of climate change and food security does not need analysis here[43]. However, we emphasize that focus group deliberations reflect expert assessments. Tensions over food versus fuel trade-offs in the development of the biofuel sector[49] are replicated in the discussion of using land for carbon stocks or bioenergy feedstocks, with implications for land tenure in global South agrarian communities and multi-functional land-use[50], as well as transnational food or bioenergy supply chains[19,51].

Groups also highlighted a desire for urban initiatives for biogenic approaches, co-benefits for health and air pollution, and public-private partnerships for green city-planning. This may be a space to watch, with sub-state initiatives to contribute Net Zero commitments are growing[1]. However, work on the types, scale, and calculability of carbon drawdown in urban settings is only beginning, as well as how to incentivize and sustain such spaces[52].

DACCS served as an archetype of high-cost, high-energy, and (potentially) hazardous infrastructure across borders. Common concerns over safety and leakage strongly reflects the public perceptions literature[14,21,33,46,53,54]. We should be wary of dismissing leakage concerns as NIMBYism. Wind turbines can be a red herring – social acceptance and opposition is tied up not only by proximity, but by the kind of infrastructure or system component, locality and vulnerability, and trust in governing institutions[21,54,55].

The issue of leakage has been recently downplayed, but the envisioned scale of future storage capacity demands caution[56]. Offshore sites possess the highest degree of safety, but leakage from storage is considered plausible due to a variety of geological and infrastructural factors, and is more likely from transport infrastructures. Assessment of effects on groundwater and soil are ongoing, and assessment gaps persist regarding sites, capacities, and impacts in global South countries[57]. Hazardous siting, moreover, is a well-documented phenomenon[58], as are the international dimensions of pollution exports and traffic in hazardous wastes[59]. These concerns are analogous to public perceptions literatures on CCS[20,24,53] and fracking[23,60]. Planning the co-location of DACCS energy sources, storage locations, and transportation will need to account for these issues.

Another concern questioned if many green technologies have hidden carbon costs, and the same fossil fuels industry perceived as culpable for climate change might be needed to power carbon capture. This reflects the public perceptions literature[18,20,53,61], as well as expert and policy debate over the energy economy of DACCS, with archetypes emerging of large-scale, natural gas-powered systems versus smaller, locale-specific prototypes with a wider potential range of energy inputs[62]. Some experts advocate for leveraging fossil fuel industries[63], but others warn of perpetuating them – for example, through enhanced oil recovery[31]. Development of renewable-powered DACCS is nascent. Policy emerging in the US plans for large-scale regional hubs that include storage and transport infrastructure, while both the US and EU are leveraging innovation funds, along with public procurement schemes being developed in the US[36]. But there is little documentation of similar plans or policies in major emerging economies. It is yet unclear how these might shift the energy and innovation economy of DACCS.

There was much less agency regarding EW, BECCS and DACCS, compared with biogenic approaches. Focus groups engaged more with DACCS than with EW and BECCS. It is plausible that the concept of ambient $CO_2$ capture technology (e.g. as artificial trees/forests) was easier to grasp in principle than the processes of weathering[20,64,65], or the coupling of bioenergy and carbon capture components[19,20].

Nevertheless, groups saw DACCS, EW (as repurposing mining waste), and BECCS more as centralized, supply-driven partnerships between government and industry, in which public participation in operationalization plays a lesser role. Sense-making of EW and BECCS also relied on familiarity with the relevant extractive or production sector, and perceptions of local-to-national co-benefits. For BECCS, these surrounded food security versus energy security issues known to bioenergy debates. For EW, these depended on perceived co-benefits from mining or using waste materials versus the energy and resource costs. The ubiquitous discussion on synergies or trade-offs with food crops and land ownership extended into both.

These speak to the need for further inquiry into the politics surrounding nationally resonant sectors of production. For BECCS, Brazil, India, and Indonesia are emerging economies with large bioenergy sectors and well-documented pressures on land-use[66,67]. EW has some antecedent knowledge to draw on through its mining or agricultural intersections. Mining requirements, the capacity for waste materials to substitute for new mining activities, and (international) transportation and supply chain logistics, are undergoing early assessments; each may undercut carbon drawdown[68]. The geopolitics of these logistics are even less parsed[69]. However, given the relevance of the agricultural sector and food security in making sense of most carbon removal proposals, investigations on EW as a fertilizer replacement and co-benefits for food production may prove influential – particularly in the global South[70].

Will carbon removal distract from decarbonization? Our results confirm a key insight from the public perceptions literature: participants

across the global North and South commonly questioned carbon removal's desirability in the context of the underlying need to reduce emissions and unsustainable resource use[20,21,25,26]. However, global South groups less commonly questioned carbon removal as a distraction from decarbonization, or mitigation deterrence[31]. Global South participants noted contextualizing circumstances: the lack of policy or media engagement with mitigation strategies, and lesser societal deliberation over climate change in general. Mitigation deterrence is also a conversation that emerged in the global North, and there may have been more travel into public discourse. Finally, while European groups tended to be precautionary in tenor, groups from key emerging economies (most clearly, China, India, Saudi Arabia) tended to be optimistic about their countries' industrial and innovation capacities.

The public perceptions literature – largely focused on the global North – is not united on the significance of mitigation deterrence. Some studies argue that it does not play a determining role in public expectations[18]; others highlight that publics acknowledge it as a concern[20,27], though more strongly if forewarned[21]. But we caution – agreeing with many[30,31,45] – that the potential for mitigation deterrence is less a function of individual preferences than of governmental and industry planning and actions. It is as important to treat the mitigation deterrence question through targeted engagement with governmental, corporate, and industry planning in key country contexts.

Publics recognize the latent roles and diversity of industry, both positively and negatively. Our findings demonstrate widespread, North-South recognition of the significance of corporate and industrial actors in innovating and upscaling carbon removal, reflected in the public perceptions literature[29,54]. These validate a prominent emphasis on policy and incentives for innovation in technological niches, but also speak to the need for clear policy and public engagement to guarantee benefits for workers and local communities[71]. Indeed, focus groups argued that assessment should be free of (hidden) political or profit motives, foster multi-disciplinary and sectoral collaboration in assessment and innovation (often tailored to local-to-national circumstances), and focus on both technical and societal appraisal (credible public consultation), often described as part of learning-by-doing or demonstration projects across biogenic and engineered/chemical approaches that clearly establish technical viability and socioeconomic benefits.

Moreover, the politics of innovation and upscaling deserve greater scrutiny. Locally and nationally determined degrees of trust in industry and government collaborations conditioned how groups envisioned private sector action, and this forms a key area for future assessment tailored to national circumstances. For groups from countries with state enterprises perceived to be reputable and successful (e.g. Norway, India, China, Saudi Arabia), there were hopes of leveraging intellectual and material resources, guided by corporate social responsibility.

However, other groups across the global North and South voiced concerns about profit-seeking and greenwashing undercutting storage permanence and safety or socioeconomic co-benefits (citing experiences with extractive industry or afforestation projects), or reflecting unequal technological capacities and potential for exporting carbon for storage (especially regarding DACCS, but implicating biogenic carbon stocks or enhanced weathering in tropical countries).

These concerns reflect numerous warnings from the expert literature: feeding hype and false expectations[72], justifying delay of industrial decarbonization[73], generating precarious local economies around carbon stocks[22], and perversely shaping government support[74]. There is a need to evaluate how corporate actors are constructing commitments towards Net Zero[1], potentially conflicting certification schemes[75], environmental, social, and industry disclosure standards, more rigorous regulations tailored to national circumstances, and transnational issues of carbon exports and technology transfer.

Agency is a key motivating factor for publics. Agency linked widespread preferences for biogenic approaches in diverse settings, trust in local governance processes, and a tendency towards self-action and community action to wrestle with 'wicked' problems. On one hand, making sense of carbon removal deployment through personal actions such as recycling recalls 'environmental individualization'[76], in which the transfer of responsibility from collective to individual action dilutes societal capacity to address complex problems. It is clear that a plurality of participants was reflexively concerned that 'demand-side' citizen actions and consumer behaviour changes should not relieve the 'supply-side' of governmental and industry action. This finding resonates with studies discussing public perceptions of responsibility for climate action, with some valuing individual level efforts such as behaviour change but others valuing structural efforts to change industrial patterns[77,78].

On the other hand, our results demonstrate a public tendency towards 'soft' pathways of carbon removal upscaling and governance: diverse, distributed systems tailored to locale, where carbon draw-down supplements developmental co-benefits, and citizen action plays a leading role in deployment[47,79]. Groups most readily conceived of kinds of ecosystems management in this vein, but one can question how smaller-scale, renewables-driven direct air capture systems might be viewed. If policy leverages this, there will be a need to match carbon accounting to a multiplicity of initiatives with diverse co-benefits, supply chains, and complex life cycles[80] – but there will be an accompanying potential for false accounting[38].

Certainly, policy will make room for 'hard' pathways, which are top-down in governance and innovation, driven by scale, optimization of resource inputs, supply chains, and storage capacities, and the calculability of carbon captured[79]. Groups did not explicitly speak to biogenic approaches in this matter, but there will be a further need to assess large agribusinesses, as well as fast-growing monoculture afforestation initiatives. Groups did, however, see DACCS, BECCS, and (the mining component of) EW systems as part of hard pathways. In our results, top-down, scale-driven systems did not necessarily lead to opposition. But in combination with technical uncertainty, perceptions of hard pathways contributed to a fear of impotence in the face of possible costs and siting impositions, and spoke to the need for engagement with the sectors and industries most relevant to siting impacts and costs versus jobs and developmental benefits.

Finally, publics sought clear coordination in (inter)governmental action. Focus groups had varying degrees of trust and understanding of governmental functions. Trust in national government and capacity varied according to a variety of anecdotal reflections – demanding more 'vertical' analyses of individual or more targeted groupings of (sub)national interests and capacities[30].

High trust in local governance can be connected to self- or community-based agency (e.g. soft pathways), as well as familiarity with public information provision and administration. Subsidiarity – action devolved to the most local level possible – would appear to be a justifiable governance principle in operationalization. However, this is half the equation, with strong support for systemic oversight and intergovernmental coordination. International policy coordination – and linking this to local action – poses a considerable assessment and policy gap, in need of advancing beyond carbon pricing and accounting towards leveraging local and sectoral co-benefits[29,79,80]. Only the EU is constructing a framework for regional, multi-lateral collaboration, and even so relies on evolving national demands[36]. At the least, there is a need to assess the evolving guidance of the Paris Agreement (Articles 6.2 and 6.4) on how carbon allowances might be transferred between countries or actors[81]. Prospects on transnational supply chains[19] or global financing[82] are the subject of nascent study. Further questions include how carbon removal presents opportunities to increase the carbon budget, and thereby redistribute 'fair shares' of emissions reductions, financing, and compensation between global

North and global South countries[83,84], or negotiating the differences internationally between subsistence versus luxury emissions, as well as residual or hard-to-abate emissions[39].

In closing, we highlight four societal expectations for informing assessment and policy that we believe to be globally robust, justified by widespread mention across focus groups in the global North and South (Fig. 1). This follows our intent to derive a global benchmark of public perceptions for informing decision-making Publics cannot be expected to speak of policy mechanisms or governance institutions with the same detail as experts or decision-makers. Rather, groups pinpoint rationales for guiding policy, or archetypes of local, national, or international mechanisms and institutions. We hope that future deliberative engagements will elaborate on public perceptions, preferences, and ensuing governance as they apply to more situated or locale-specific contexts: for example, regional portfolios of carbon removal, and demographics particularly in the global South.

Assessment and decision-making should prioritize public engagement, going beyond acceptance research that treats publics as consumers of carbon credits or products, or as adopters for whom carbon removal must be 'de-risked'. A move towards meaningful consultation can be nuanced by kind of carbon removal. Pilots and demonstration projects for bottom-up, distributed land-use or marine ecosystems management approaches could treat publics as fruitful drivers of upscaling, and emphasize mutual learning with triple helix actors[85]. But all biogenic and engineered approaches raised public concerns – from navigating spatial trade-offs to potentially hazardous facilities or transportation networks – where consultations would be valuable for establishing adherence to social and environmental criteria and local co-benefits. This would work towards widely held public imperatives of agency, of those affected being involved in governance, and of tendencies towards trust in local governance processes.

Scrutiny and regulation of the role of industry in carbon removal should be developed beyond incentivizing innovation. Focus groups often raised the Polluter Pays principle (PPP) as a call for polluting industries to pay for high-cost, high-energy options such as DACCS (similarly debated by experts[82]), and highlighted a role for a role for companies and industries to marshal their resources for upscaling biogenic approaches – for example, through tree planting campaigns. Moreover, surrounding deliberations show that the PPP discursively reflected a range of wider concerns regarding industry and corporate agendas – which applied not only to DACCS, but to biogenic approaches such as forestry management. In other words, publics discussed the PPP more expansively as 'polluters should take responsibility'. Concerns included industry-governmental collusion and corruption, profit-seeking motives undercutting local co-benefits (all approaches) and carbon storage safety and permanence (leakage issues for DACCS and BECCS; deforestation pressures in forestry management), and concerns that costs and harms should not be passed to citizens (e.g. hazardous siting of DACCS infrastructure, land-use trade-offs for afforestation or bioenergy feedstocks for BECCS, and land/property appropriations for all approaches) or to other countries (pollution exports of stored carbon). For all approaches, decision-making should be wary of fossil fuel, extractive, and agribusiness interests co-opting or diverting policy – attempting to overclaim 'residual' or 'hard to abate' emissions, cosmetically rebranding existing activities as carbon removal (e.g. land-use practices), or using development of carbon removal to greenwash industry expansion (e.g. DACCS in the fossil fuel industry). These risks are relevant even in countries with positive perspectives on state-industry collaborations and corporate social responsibility.

Systemic coordination of carbon removal portfolios and policy must be a key (inter)government function. Coherent national strategies

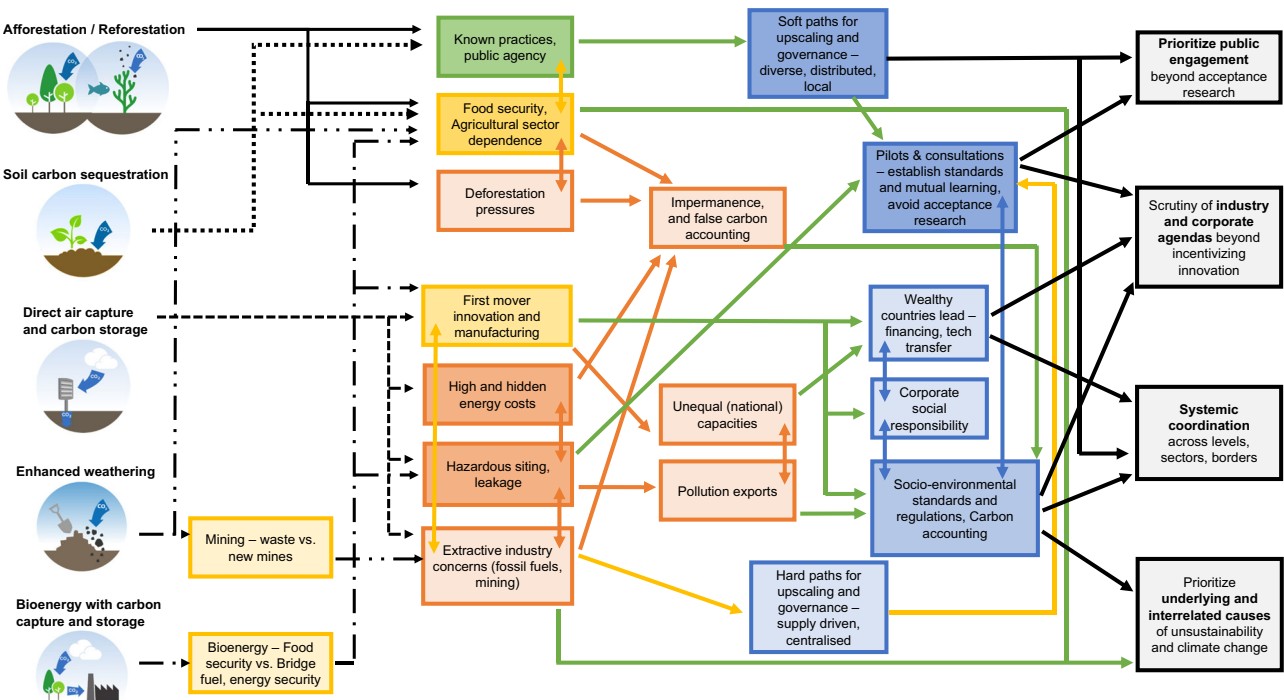

**Fig. 1 | Complexes of hopes, concerns, and corresponding governance issues associated with carbon removal.** To the left are the five archetypes of carbon removal. The pictures are those provided in information materials sent to participants. The carbon removal types are connected to hopes (in green), concerns (in orange), and themes with elements of both (in yellow). Governance rationales and activities are in blue. The arrows signify linkages between hopes, concerns and corresponding governance; the colours of the arrows correspond to the aforementioned scheme – positively, as a hope (green); negatively, as a concern (orange), both (yellow), and connections between governance (blue). The darker the colour of the boxes, the more focus groups spoke to the theme contained therein. These complexes connect to one or more of the four robust societal expectations for informing governance in the conclusion, outlined in the black blocks to the right.

for multi-level, sectoral, and regional coordination will certainly be a first and logical activity, which mirrors their emergence in the US, and even regionally at the EU. However, all countries must foreground the latent transboundary dimensions of carbon removal. At a broad level across carbon removal approaches, these include: (re)evaluating 'fair shares' of the carbon budget, distributing different kinds of carbon removal and components of international supply chains, separating targets for emissions reductions and carbon removal[86], clarifying definitions of hard-to-abate and other categories of emissions that implicate offsets[39], and separating biogenic carbon removal from offsetting industry emissions[38] – all in light of the potential for bilateral and multilateral trading of carbon credits. There are further nuances across approaches. The agriculture, forestry and land-use sector is disproportionately significant in tropical forested countries – and Latin and South America further contains the touchstone issue of Amazon governance. Governments must negotiate the dimensions of using these regions to generate credits from carbon forestry or bioenergy feedstocks for BECCS. For options such as DACCS, both global North countries and emerging economies with the technological capacity will need to develop processes for financing and technology transfer, while negotiating the transboundary movement of carbon to offshore or land-based reservoirs.

Finally, both researchers and policymakers must prioritize underlying and interrelated causes of unsustainability and climate change. Certainly, the demand that carbon removal must not be allowed to become a delaying tactic through mitigation deterrence is much repeated – even so, incentives for delay remain resilient[45,87]. Furthermore, carbon removal should not be treated as a technocratic carbon management strategy. The management of agriculture and forestry as carbon stocks – linking further to diverse ecosystems management practices – implicates food security and biodiversity, as is increasingly recognized in syntheses assessments[43]. The dimensions are also true of the bioenergy component of BECCS, or the use of enhanced weathering as a soil additive. DACCS implicates energy governance, security, and accessibility, as well as corporate governance, given its potential to be powered by incumbent fossil fuel expansion or expanded renewable energy development. The transportation and storage of carbon from DACCS or BECCS covers a range of logistics and locales that extend from land-based reservoirs to the ocean floor, in turn implicating a range of governance architectures. Versions of biogenic and engineered approaches may well be incorporated into urban planning, given city-based commitments towards Net Zero. Carbon removal development may be an opportunity to integrate the governance of entwined global issues.

## Methods

### Inclusion and Ethics Statement
All components of the research were granted ethical approval by the Research Ethics Committee of Aarhus University (#2021-13). Full and informed consent was given by all participants before the beginning of the study, along with all participants being notified about the fact that their data would be handled in a fully anonymous manner and in complete accordance with the General Data Protection Regulation and any other pertinent data-security regulations, that any data would be analyzed in an aggregate fashion and would not be personally identifiable in any way, and that they had the right to withdraw their participation at any time. The research has been broadly undertaken with the aim of better understanding public perceptions of carbon removal approaches, including in the Global South and by means of more qualitative methods that can better elucidate the variability and importance of the local context. At this stage, no local researchers have been included. The specific roles and responsibilities of those in the author team was discussed prior to the research. Insofar as possible, we have striven to take into account local and regional research in the citations.

### Project and mixed methods framework
Our study is part of the project Geoengineering and Negative Emissions Pathways in Europe (GENIE), a multi-institutional assessment of carbon removal and solar geoengineering approaches funded by the European Research Council. The following research framework forms the basis for several emerging publications and other contributions. We utilize a multi-methods framework combining a survey instrument with focus groups. The survey's design (technological and geographic scope, methodology, materials) is described in Baum et al.[88]. The focus groups form the basis of this paper, and the study design is described below. We foreground that focus groups – reflecting small sample sizes, screening for particular characteristics, and potentially driven by emergent topics and dynamics particular to each group – should not be seen as (wholly) representative of a national public. Nationally representative results are the province of large-N surveys[88]. Here, we aim at the in-depth treatment and detail that comes with deliberation, and comparison with the wider public perceptions literature. Furthermore, focus groups frequently refer to their local or national contexts – which is significant and can form the basis for further situated assessment, even if not definitively representative. In combination, the survey sources more aggregate, nationally-representative preferences, while the focus groups trace rationales and processes.

### Choice of carbon removal options
We chose five rough (arche)types of carbon removal that represent a representative spread of biogenic approaches to chemical or engineered approaches (IPCC, 2022 Chapter 12, Box 8, Fig. 1). We chose two biogenic (sets of) approaches: (1) afforestation and reforestation, which was treated as 'restoring and/or growing vegetation' as an entry into wider and more diverse practices in managing terrestrial and marine ecosystems, including blue carbon, and (2) soil carbon sequestration – including, but not interchangeable with biochar as a related approach for fixing carbon in agricultural management practices. We did not engage with biochar's other potential applications – e.g. in construction. We then chose two distinct (types of) chemical approaches: (3) direct air capture and carbon storage (DACCS) and (4) enhanced weathering. Finally, we opted for (5) bioenergy carbon capture and storage (BECCS), a hybrid system that combined a bioenergy input (and therefore, a land-use component with overlaps to biogenic approaches) with a technical storage component (linking to DACCS).

However, we foreground that these choices elided nuances within, and links between, each rough type. Assessment is disproportionately growing on biochar as a potentially widespread, customizable option; our study design includes this only within general soil carbon sequestration and agricultural management practices. Ocean alkalinization, a large-scale, marine variant of enhanced weathering that has distinct socio-technical characteristics in comparison with more distributed conceptions of terrestrial enhanced weathering, is elided. In general, marine based carbon removal approaches were noted but not clearly highlighted in our materials and questioning.

### Solar geoengineering
Half of the focus group run-time are given to solar geoengineering approaches (and the other half to carbon removal). These two suites of approaches have historically been grouped together under the umbrella concept of (climate) geoengineering, but debate on the conditions under which they should be separately (representing different socio-technical characteristics and governance demands) or comparatively assessed (for synergies and trade-offs in the context of wider climate action) remains inconclusive. In this paper, we focus on carbon removal in order to maintain topical coherence. A twinned paper assessing public perceptions on solar geoengineering, deploying the same framework, is in preparation. We foreground a well-

known comparative effect: that assessing carbon removal and solar geoengineering side-by-side pushes perceptions in favour of carbon removal options, and in particular, biogenic (so-called 'nature-based') options. Although we do not report this – due to our choice to separate between carbon removal and solar geoengineering in different outputs – our data confirms perspectives that negatively compare technically-uncertain (e.g. by references to 'science fiction') forms of planetary solar geoengineering against low-cost, decentralized, technically-graspable forms of carbon removal. However, we also find evidence that (certain) carbon removal and (all) solar geoengineering approaches are both viewed as stopgap solutions that do not address the root causes of climate change.

## Country selection

Forty-four focus groups (1 urban, 1 rural) were conducted in 22 countries worldwide, aiming at a roughly even split between countries in the global North and global South, geographic spread across all UN regional groups, and inclusion of regional powers. A prior stage deployed a survey instrument on carbon removal approaches in 30 countries – all 22 assessed here, as well as Canada, France, the Netherlands, Norway, Estonia, Greece, Denmark, Singapore, and Japan[88]. Constrained resources led to a refinement of scope and reduction of countries in which focus groups could be run. The countries and groupings used for the focus groups are: Global North (11 countries): US, UK, Australia, Germany, Austria, Switzerland, Italy, Poland, Norway, Sweden, Spain; Emerging South (8 countries): South Africa, India, Chile, Brazil, China, Turkey, Saudi Arabia, Indonesia; and Developing (3 countries): Kenya, Nigeria, Dominican Republic. We use the rough distinction between emerging and developing economies used by the International Monetary Fund and its World Economic Outlook reports[89] (between the emerging market and middle-income economies and low-income developing countries) to imperfectly acknowledge intra-South differences.

## Participation and recruitment

Recruitment aimed at 8 participants per focus group. Due to technical difficulties and dropouts, the lowest number of participants was 5. The total number of participants was 323. Recruitment was conducted in collaboration with Norstat, a European-based data collection company (https://norstatgroup.com/).

Prospective participants were screened via an online survey for a number of mandatory and 'soft' criteria. Climate denialism was screened out (all who answered "No" to "Do you believe climate change is happening?"). Focus groups were further screened for an even split between genders, and between young (18–44) and old (above 45+) cohorts. Participants were screened for an urban (including suburban) or rural background, which was self-defined, and relied on responding "Urban", "Suburban", or "Rural" to the question: "How would you describe the area in which you live?".

Two further guiding but not mandatory screens were held. The first was for distribution across education level, income, and occupation type, each tailored by country. The second was for distribution across regions within a country. In most cases, these were defined by formal (e.g federal) administrative regions or states/provinces; in a smaller number, these were defined by broad geographic regions (in USA, India, Brazil, Indonesia).

## Materials and languages

Two sets of materials – a discussion guide (of questions, topical emphases, and timings) for moderators, and information materials on approaches (distributed to public participants beforehand) – were developed primarily by the authors, in collaboration with Norstat. Materials were written originally in English and communicated in that language with focus groups in US, UK, Kenya, Nigeria, South Africa, Australia, and India. Materials for other countries were translated into:

German (Germany, Austria, Switzerland); Italian (Italy); Polish (Poland); Norwegian (Norway); Swedish (Sweden); Spanish (Spain, Chile, Dominican Republic); Portuguese (Brazil); Mandarin Chinese (China); Turkish (Turkey); Arabic (Saudi Arabia), and Bahasa Indonesia (Indonesia). All technical terms were translated from English into their native languages by academic experts (all colleagues in climate and energy governance known to the authors).

## Discussion guide

This consisted of the following questions, in four groupings, as follows. The guiding logic was to focus conversation on actors, actions, and agendas at the most tangible scale possible.

The first grouping of questions was based on hopes, or prospective benefits. The questions were as follows. What are the benefits from any of these approaches? Who might gain the most from these benefits, and why? If these were implemented in your community or country, who would be affected positively – and how and why?

The second grouping of questions was based on concerns, or prospective risks. The questions were as follows. What are the risks from any of these approaches? Who might be most negatively impacted from these risks, and why? If these were implemented in your community or country, who would be affected negatively – and how and why?

The third grouping of questions was based on corresponding governance. The questions were as follows. In an ideal world who are the most significant people that should help make decisions on this approach – in your community, or your country, or even the world? What actions should be taken before there is consideration to implement this approach – what would you like to see done? How would you want yourself, and the wider public, to be involved in making decisions on these approaches?

Finally, a 'headlines exercise' was conducted: A creative mini-scenario exercise was held, in which participants were asked to create a (newspaper) headline in 2030, with four elements: an approach, an actor, and an event, in sum representing a good or bad outcome related to the approach (a headline that makes the participant feel hopeful or worried).

## Informational materials on approaches

The research design did not seek to capture spontaneous views on carbon removal, and involved a 'learning phase' involving the prior dissemination of informational materials. This step was taken to account for the limited discussion time of each focus group, the lack of resources to conduct a more concerted reflection and questioning with technical experts as part of focus groups, and the fact that there is a lack of directly lived experience we regard to certain approaches (although there clearly are analogous lived experiences).

Information materials were sent to participants a week prior to the conduct of the focus group. Participants were encouraged to do further research, and to discuss with family, friends, and members of their community[20]. The information materials consisted of the following elements. There was one introductory page each for carbon removal and solar geoengineering as separate suites, with bullet points on their overarching characteristics. The carbon removal introductory page emphasized a spectrum between biogenic (afforestation and reforestation; soil carbon sequestration), to chemical/engineered (DACCS, enhanced weathering), to a mix thereof (BECCS).

The materials were careful not to use the term "nature-based", understanding that the term has documented steering effects[34]. Instead, materials referred to these as approaches "that change how we use nature". Chemical/engineered systems were referred to as utilizing "large-scale engineering systems". In following pages, each approach was accompanied by a column of approach-specific text, taking up one-third to half a page each. Each column contained: a brief technical

description; a 'cartoon' picture, deliberately stylized to avoid reification; a short list of technical infrastructural needs; and a point or two each of key technical pros and cons that were extremely abbreviated to forestall as much framing as possible.

## Meeting logistics

The majority of meetings were conducted online, via Zoom (version 5.17.7 (31859)), which we selected for ease of logistics, costs, recording, and transcription. Meetings in Dominican Republic, Nigeria, Kenya, South Africa, and the rural group for India were held in person or in hybrid format. Meetings were moderated by Norstat personnel, in the same language in which materials were translated into (see Materials for the list of native languages). All focus groups ran for at least 2 h, with the carbon removal and solar geoengineering suites each receiving half the allotted time. Half of the focus groups began by discussing carbon removal, and the other half with solar geoengineering. We recognize that the time allotted was slim, relative to other deliberative exercises. This was due to compromise between our financial resources, and the inclination of our research design towards greater geographic coverage as part of a global, horizontal benchmark of hopes, concerns, and expectations of assessment and governance for further deliberation.

## Transcription

Online groups were recorded via Zoom. For hybrid and in-person groups, various other recording mediums were used. All deliberations were transcribed by Norstat. All transcripts went through multiple rounds of clarification between the transcribers and the authors to ensure accuracy and quality.

## Coding and analysis

The authors conducted a two-part analysis. The first was 'horizontal': using qualitative data analysis software MaxQDA to code (variants of) themes across focus groups and countries. The primary coders from the author team were SL and LF, using qualitative data analysis software MaxQDA (MAXQDA Standard 2022, Release 22.8.0, (c) 1995-2022 VERBI GmbH Berlin). SL coded all urban groups, while LF coded all rural groups; both coders conducted frequent checks to establish reliability. Initial macro-coding was organized according to the questions (see discussion guide in Materials), cross-referencing individual or collective carbon removal approaches with perspectives on the following: (a) Climate change causes and impacts; (b) Benefits and 'Winners'; (c) Risks and 'Losers'; (d) Governance; and Publics, with further coding emerging on (f) Contexts and analogies and (g) Technical uncertainties. The second was 'vertical': writing more qualitative deep-dives into each focus group's emphases, contestations, and agreements; then combining the urban and rural summaries into country-by-country summaries. Again, LF analyzed all rural groups; SL analyzed all urban groups. The key technical and societal issues presented in the results section are derived from the bottom-up coding. In our results, we foreground where results agree or diverge with the public engagement literature. In the discussion, we further compare our results against expert assessment (beyond public engagement), innovation, and policy at a landscaping level.

## Urban versus rural

Due to a need to refine the focus of this paper, we chose not to undertake a deep investigation of the differences between perspectives of urban versus rural groups cutting across approaches and countries. During a preliminary analysis, the authors found that in most countries, any topical differences and nuances between urban and rural groups reflected differences in emphases rather than reflecting opposition. The differences are complementary, rather than contradictory. In certain countries, urban and rural groups touched upon all topics in almost identical ways. It was not even clear

from our preliminary analysis if rural groups spoke with greater preference and detail to soil carbon sequestration and other biogenic approaches and components (forestry management, bioenergy) – both urban and rural groups spoke to these approaches in comparable detail, and there were no immediately recognizable, fundamental differences in content or inclination. Rather, participants in both rural and urban groups demonstrated mutual understanding of contexts and local issues, and attempted to gauge and discuss how carbon removal approaches would be helpful in both urban and rural contexts: e.g. through portfolios of approaches tailored to geography and context, and questioning different kinds of siting and pollution dumping issues.

Without specific and detailed analysis, it is difficult to foreground why these (initial, non-definitive) dimensions emerged. Numerous contexts are plausible: that participants have personal and professional networks that stretch beyond their immediate locales; that they have themselves moved between urban and rural locales (to which participants across many countries admitted or alluded to); that national and international print and online media play a role; and that people can – intuitively, or with deliberation – think richly in terms beyond the tangibly local. For now, we must reserve further inquiry on this topic to a later endeavour.

## Reporting summary

Further information on research design is available in the Nature Portfolio Reporting Summary linked to this article.

## Data availability

The datasets generated during the current study are available upon request, subject to licensing agreements and ongoing research of the project (European Research Council Grant Agreement No. 951542-GENIE-ERC-2020-SyG).

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

## Acknowledgements

This project has received funding from the European Union's Horizon 2020 research and innovation programme under the European Research Council (ERC) Grant Agreement No. 951542-GENIE-ERC-2020-SyG, "GeoEngineering and NegatIve Emissions pathways in Europe" (GENIE). The content of this deliverable does not reflect the official opinion of the European Union. Responsibility for the information and views expressed herein lies entirely with the author(s). The authors acknowledge the numerous colleagues who helped translate key terms into different languages for the information materials; Norstat personnel for organizing and moderating the focus groups; and D.P. McLaren and E. Cox for technical clarifications and comments on the interview questions and discussion guide.

## Author contributions

S.L., L.F., C.M.B and B.K.S. designed the study. S.L and L.F. undertook data analysis and synthesis, with input from C.M.B. S.L. wrote the first draft of the manuscript, with content and reference inputs from L.F. and C.M.B. S.L., L.F., C.M.B and B.K.S. wrote and edited the manuscript to completion.

## Competing interests

The authors declare no competing interests.

 
