## [Peer Review File · Nature Communications]

Public perceptions on carbon removal from focus groups in 22 countriesREVIEWER COMMENTS

Reviewer #1 (Remarks to the Author):

This manuscript covers a tremendous amount of ground, and for the most part does it quite well. I have two significant criticisms, however, one more substantive and one more stylistic.

First, the paper's conclusions feel underwhelming, especially given all the work that precedes them. The four governance principles you identify—prioritize public engagement, polluter pays, systemic coordination, prioritize root causes—are certainly robust, but provide relatively little insight into how particular types of CDR should be governed. Much of this has to do with attempting to derive principles applicable to all CDR approaches—the results are so rich and varied that any principles formulated to connect them end up being rather generic, while the connections between any principle and specific results claimed to embody it are often unclear. This is a case where lumping all CDR approaches together comes at the cost of sharp analysis.

For example, you write “The Polluter Pays principle is a call for polluting industries to pay for high-cost, high-energy options (e.g. Honegger, 2023) – and more. Surrounding deliberations show that it is a catchall for concerns regarding industry and corporate agendas – driving governmental collusion and corruption, profit-seeking motives undercutting benefits and carbon storage safety and permanence, and concerns that costs and harms should not be passed to citizens (e.g. taxes, or hazardous siting) or to other countries (pollution exports).” To argue that the PPP somehow includes mitigation deterrence, concerns about permanence, and other worries strikes me as incoherent—these are fundamentally different issues, ideas, and risks. The PPP does in fact seem particularly applicable to “high-cost” options, i.e., technological/industrial/engineering CDR like DACCS, EW/OAE, and BECCS. The source of the incoherence lies in trying to stretch the PPP to cover natural/nature-based/biogenic CDR as well, which leads for instance to trying to shoehorn the permanence issue into the PPP.

A better approach would be to seek to derive 1) governance principles applicable to technological CDR alongside 2) governance principles applicable to nature-based CDR, which may in some cases overlap. You already repeatedly note the significant differences between these two categories, and real-world politics is starting to fracture along those lines as well. Natural and technological CDR vary in important ways including cost, maturity, risks, co-benefits, permanence, and removal potential—certainly repeating the exercise illustrated in Figure 2 at the level of these two types of carbon removal will “uncover” governance principles that are less generic, more specific, more appropriate, and more policy-relevant. I strongly encourage you to shift your level of analysis down one level from all CDR to natural CDR and technological CDR side by side in the conclusion—I am confident this will lead to a bigger payoff.

Second, the tendency to mix statements made by focus groups together with what appear to be the authors' own inferences based on those statements (including extensions by proxy), in ways that seem to attribute the authors' understandings to focus group participants, is a persistent problem. Several examples are discussed below. The text needs to be revised to clearly separate presentation of results in terms of participant observations from discussion of those results including their implications based on the authors' analytical framework.

More specific comments follow.

Figure 1—"Countries in which focus groups were represented" include only colored (non-gray) countries—is that correct? If so that needs to be stated explicitly because it's not self-evident. It may be clearer to use a table instead.

Afforestation and reforestation—Throughout this section you write in terms of focus groups discussing, supporting, and/or opposing "ecosystems management," but from what I can tell—including based on reading the Methods section—these groups focused specifically on A/R, not on ecosystems management more broadly, while the authors treated A/R as a proxy for ecosystems management. If I understand this correctly, then it is important to revise this section to make clear that focus groups talked about A/R and that the authors are treating this as a proxy for ecosystems management, and that focus groups did not and were not instructed to discuss the more general category ecosystems management.

Line 214—This section reads as though the distinction between A/R (as CDR) and stopping deforestation (as emissions reduction) was unclear in focus groups—if so, that seems worth noting. Could participants lumping these two linked but separate approaches together have affected their deliberations?

Soil carbon sequestration—Similar to the issues surrounding A/R, this section is written as though focus groups explicitly discussed biochar, when in fact they discussed SCS without specific reference to biochar while the authors treated SCS as interchangeable with biochar. If this is correct, then the text here needs to be revised accordingly to avoid giving the impression that focus groups deliberated on biochar.

Line 406—When speaking about "the underlying causes of climate change," it is unclear whether the authors mean causes as identified by focus groups, causes as identified in the literature, and/or causes as identified by the authors themselves. This seems to continue the pattern of appearing to conflate the views of participants with the views of the researchers.

Line 437—Surely participants did not explicitly reference the triple helix model but rather what researchers refer to as the triple helix model?

Line 608—You seem to suggest that the downside of preference for natural/nature-based CDR lies in its potential substitution for emissions cuts. But aren't its limited removal potential and nonpermanence/vulnerability to re-emission equally significant downsides?

Reviewer #2 (Remarks to the Author):

This manuscript reports results from 44 focus groups addressing carbon removal, broken down by urban/rural distinctions in 22 countries. The authors develop a “robust” approach, examining prominent themes that emerged in the focus groups against findings from wider literatures on perceptions of carbon removal. In their words, this approach aims “to see how a broadened set of publics nuance ... prevalent findings”. In each focus group, they examined 5 carbon removal techniques starting with hopes and concerns before moving to governance questions. The reporting of the Results mirrors this research design, a sub-section given to the 5 technologies and sub-section on cross-cutting governance dimensions, with a comparatively brief section on carbon removal and climate/sustainability action sandwiched in the middle. The Discussion mirrors the three Results sections and draws out some of the wider implications. In the Conclusion the authors outline four “robust governance principles” and provide a diagram linking these to the various results reported.

This is an admirably ambitious cross-cultural study on perceptions of carbon removal. The authors are reflexive about the analytical challenges that such a research design poses and, in general, their “robust” approach is well justified. Overall, the paper has potential to make an important contribution. However, currently the positioning of study and definition of its scope and aims are quite loosely defined. This is reflected in the title of the paper, which simply describes the research design rather than summarising the study's contribution. The Research Design section states that the aim of the analysis is to construct a global “benchmark” for deliberative research on carbon removal- this sounds like quite an important contribution and I wondered if it could be introduced earlier and perhaps elaborated? Clarifying the scope/aims would help the reader follow the analysis which, although containing many insights, sometimes seems to jump between sections and subsections without a clear connecting narrative.

One effect of the authors' pluralist-aggregative analysis, is that in presenting the Results it's not always clear when they are reporting participants' own words and when they are applying concepts (particularly in the Governance sub-section). I appreciate that this approach necessarily requires the reduction of participants' speech, but currently they only speak in their own words when they're illustrating the themes of your coding (in the boxes). If you could show a bit more of the discursive interaction and the kinds of ambivalent speech that often emerges in focus groups, this would bolster your overarching claims about deliberative research and make the study's cultural pluralism more tangible. It might also head-off criticisms of your aggregative approach as overly-reductive. Related to this, your methodology

involves both a bottom-up approach to coding and an analysis informed by themes derived from perception literatures (e.g. Smith et al, Sovacool et al). You suggest these are presented separately in the Results and Discussion, respectively. But I wasn't clear whether the "key technical and societal issues" presented in the Results section were derived from the bottom-up coding alone? Can you briefly elaborate on this, perhaps in the section on coding methodology.

More generally, the reason for structuring the Results in three sub-sections could be better justified and narrated. Sections 1 & 3 (the 5 technologies, governance issues) seem clearly reflected in the research design. It's less clear, at least from a linear reading, how the climate/sustainability action section was arrived at. To help the reader, this could perhaps be briefly elaborated in the Introduction.

The Discussion section contains many interesting insights, but (perhaps because they mirror the structure of the Results) I wasn't always clear how these related to the overarching aims of the paper. The Introduction suggests that this section is where you compare your findings against perceptions literatures and expert/policy debates. But don't you also do that to some extent in the Results section? Clarifying this further would help the reader.

The Conclusion presents "Four Robust Governance Principles". These principles are well-rehearsed in the carbon removal literature, as the authors acknowledge. In this sense they are certainly robust. But the distinctive contribution of your study seems to get a little lost here. Could you perhaps say something about the aim to construct a global benchmark for deliberative research on carbon removal?

Specific comments:

p.4 ll146-8: rather than talking about your future plans for 'vertical' analysis, perhaps you could elaborate more on the contribution the 'global, horizontal' approach can make?

p.5 l194: "Focus groups in every nation". Is nation the right term here?

p.5 l223: "This broad range of initiatives will need to be made more practical on national and local levels." Sounds odd to be drawing out learnings at this stage of the paper. Consider rephrasing.

p.5 l224: "lack of technical understanding regarding the scale and calculability of urban flora" – can you be more specific e.g. do you mean in terms of carbon sequestration?

p.6 l231: “strong global pluralities”. This means a plurality of groups across Global North/South right?

p.7 l275: “a surprisingly common concern was on soil toxicity”. Why surprising and why can it be questioned as a technical misunderstanding?

p.7 l277: “carbon leakage” – are you referring here to policy/accounting debates about this concept? Perhaps briefly clarify. (this question applies to several other points where you use the term)

p.7 ll280-2: “participants found DACCS more difficult to technically grasp, while seeing it as a centralized, supply-driven cooperation between government and industry”. Could you be more specific e.g. was it the chemical removal process?

p.7 p.302: “NIMBYism unsurprisingly emerged in global North groups”. I’m clear whether you are characterising some of your participants as NIMBYs here? This term has been much critiqued- consider using a less contentious concept?

p.8 ll334-6. As above, when you say that “technicalities” were hard to grasp it would help if you were more specific.

p.9 ll389-90. Technical misconception about “double removal”. Hasn’t double counting in BECCS supply chains also been raised as a concern by some experts?

p.12 l436: “Participants ubiquitously referenced a high degree of trust in expert-driven assessment, often referring to the ‘triple helix’ model”. I’m curious about how this was expressed, particularly in countries where science plays a less prominent role in public life.

p.18 l582: The title of the Discussion section comprises a set of quite broad themes. Could the focus be clarified?

p.21 l845: “Subsidiarity would appear to be a justifiable governance principle in operationalization...”. I didn’t follow this point.

p.22: The diagram accompanying the conclusions, though complicated, is potentially a useful roadmap for the reader- would it make sense to introduce this earlier?

p.26 l1099: Can you briefly justify the choice to use of Zoom for the focus groups instead of a more traditional face-to-face set up? I appreciate there are obvious practical and resource-related reasons for doing so, but there is also a lot of methodological discussion that you could signpost here.

p.26 l1101: If I understand right, most focus groups discussed carbon removal for around 1hour. That's not a long time at all, relative to other deliberative research on carbon removal. I appreciate you gave materials in advance and allowed participants to discuss and research the topic. Could you say something briefly to qualify this approach to deliberative research?

I'm happy for the authors to contact me directly if they would like to discuss or clarify any of the above comments.

Reviewer #3 (Remarks to the Author):

This is a useful and interesting study of public perceptions of carbon removal (CR). Its strength is in its global reach and breadth, with 44 focus groups held in 22 countries. Its findings do not necessarily present any surprises, as the authors acknowledge, given that public perceptions of CR were already well studied. However, the international focus – and particularly, applying the same methodology across all countries, is novel. The paper offers some sound advice on governance principles for CR which again are in line with similar previous work. I recommend it for publication if the issues addressed below are considered.

I would suggest two areas for revisions, as follows:

- 1) Clarity about the purpose of consultation. Why were people being asked for their views on CR? Many CR technologies are not directly consumer-facing, ie, unlike in other domains eg transport or home energy, they do not require changes in behaviour or actions. Therefore (as with other technologies eg grid infrastructure) the point of investigating public perceptions is not to influence consumer uptake. Is the aim to guide policy support for and governance of CR, as the conclusions imply? If so, this should be made more explicit in the initial parts of the paper.

At times in the paper, it seems as if the aims of the consultation were to compare public perceptions with stakeholder perceptions (or 'lay' vs 'expert' as commonly described). If this is the case, this is valuable – for example, publics may – and usually do – prioritise notions of fairness and justice, which is an important lens through which to consider governance. However, the paper also comments on how well publics 'understood' or 'grasped' the fundamental technical aspects of different CR techniques. I am not sure of the purpose of this. Given that many CR techniques are not yet commonplace, it is hardly surprising that understanding is low. At times, the paper seems to lapse into judgement of public

perceptions which are deemed to be based on inadequate knowledge, eg line 275 'a surprisingly common concern was soil toxicity' / 318 'strangely, the pollution export concern was raised in almost no...' / 792 'troublingly, there was almost no awareness...' This poses the danger of a 'deficit model' of public engagement, where the focus is on 'correcting' public perceptions through correcting a perceived information deficit. Is it an explicit aim to compare – and evaluate – publics' understandings compared to stakeholders/experts? Or is the aim more a democratic one, to ensure that governance is informed by public views and values?

2) On the methodology, the chosen method was focus groups, yet the project did not seek to capture spontaneous views on CR, but provided upfront information on different CR techniques and approaches, as detailed in the annexe. This is actually moving toward more deliberative research (which specifically involves a 'learning phase') and considerably alters the likely outcomes. I can see the reasoning for providing basic information about CR technologies, given that there is little lived experience of them, but the fact that the study involved a 'learning phase' / informational input / stimulus materials needs to be stated upfront.

Further minor points:

The use of the term 'manpower' is outdated – could the gender neutral term 'labour' be used instead?

Line 228 – limited discussion of offsetting – this is very interesting, and relevant to questions of governance – could it be explored further?

291 – perceptions of government-industry collaborations – is this influenced by deeper questions of trust in government?

305 and elsewhere – might be worth stressing that the term 'leakage' refers to actual physical leakage of stored gases (I think) – compared to the often-used phrase 'carbon leakage' in a metaphorical sense, ie imports/exports of embodied carbon in products.

402 and 625 – concerns about CR tended to be linked to localised concerns – were the groups provided with any information about global carbon budgets or the Paris-agreed target of net zero emissions? This is likely to have a significant impact on views of CR. i.e. there is no IPCC global pathway (or other significant modelling) which forecasts achievement of the net zero target without a certain amount of CR. Knowledge of this is likely to significantly influence people's attitudes toward CR (as seen, for example, in national Citizens' Assemblies on climate change. Without background understanding of carbon budgets, people may see CR as one among a range of different options for meeting net zero targets.

REVIEWER 1

This manuscript covers a tremendous amount of ground, and for the most part does it quite well. I have two significant criticisms, however, one more substantive and one more stylistic.

First, the paper's conclusions feel underwhelming, especially given all the work that precedes them. The four governance principles you identify—prioritize public engagement, polluter pays, systemic coordination, prioritize root causes—are certainly robust, but provide relatively little insight into how particular types of CDR should be governed. Much of this has to do with attempting to derive principles applicable to all CDR approaches—the results are so rich and varied that any principles formulated to connect them end up being rather generic, while the connections between any principle and specific results claimed to embody it are often unclear. This is a case where lumping all CDR approaches together comes at the cost of sharp analysis.

For example, you write “The Polluter Pays principle is a call for polluting industries to pay for high-cost, high-energy options (e.g. Honegger, 2023) – and more. Surrounding deliberations show that it is a catch-all for concerns regarding industry and corporate agendas – driving governmental collusion and corruption, profit-seeking motives undercutting benefits and carbon storage safety and permanence, and concerns that costs and harms should not be passed to citizens (e.g. taxes, or hazardous siting) or to other countries (pollution exports).” To argue that the PPP somehow includes mitigation deterrence, concerns about permanence, and other worries strikes me as incoherent—these are fundamentally different issues, ideas, and risks. The PPP does in fact seem particularly applicable to “high-cost” options, i.e., technological/industrial/engineering CDR like DACCS, EW/OAE, and BECCS. The

Thank you for this recommendation!

We had intended these principles to serve as a brief collation of the paper's content combined with a concluding call to action. Taking your recommendation requires lengthening, as well as some repetition with previous sections – but we are happy to do.

We first need to lay out some context.

In the conclusion and the paper more broadly, we were trying to navigate two directions/issues.

The first is what you highlight and recommend: that there is specificity between biogenic and engineered approaches (or nature based and technological) in “cost, maturity, risks, cobenefits, permanence, and removal potential”. Indeed, this would be true if broken down even further to particular approaches, or to variants of approaches (e.g. kinds of enhanced weathering, or sorbent vs. solvent direct air capture).

The second is based on more aggregate, broad principles that (a) emerged from complexes of hope, concern, and preferred governance that are robust across CDR approaches, though with key nuances (see Figure 2), and (b) should apply to governance irrespective of CDR approach.

We still see value in deriving principles robust across approaches (and countries) – since, in trying to derive such principles, we found that there were as many overlaps as gaps between biogenic and engineered approaches. Put another way: we tried to derive separate lists of principles – and found that they were sufficiently overlapping to warrant re-combining them.

	Nevertheless, your comment that lumping often does come at the cost of sharp(er) analysis has given us cause for reflection,
--	--

source of the incoherence lies in trying to stretch the PPP to cover natural/naturebased/biogenic CDR as well, which leads for instance to trying to shoehorn the permanence issue into the PPP. A better approach would be to seek to derive 1) governance principles applicable to technological CDR alongside 2) governance principles applicable to nature-based CDR, which may in some cases overlap. You already repeatedly note the significant differences between these two categories, and real-world politics is starting to fracture along those lines as well. Natural and technological CDR vary in important ways including cost, maturity, risks, co-benefits, permanence, and removal potential—certainly repeating the exercise illustrated in Figure 2 at the level of these two types of carbon removal will “uncover” governance principles that are less generic, more specific, more appropriate, and more policyrelevant. I strongly encourage you to shift your level of analysis down one level from all CDR to natural CDR and technological CDR side by side in the conclusion—I am confident this will lead to a bigger payoff.	and driven us to show greater specificity and nuance in the existing principles between biogenic and engineered approaches. With regard to the “polluter pays principle” (PPP) paragraph, which you highlight as an example, we have reformed it to lead with a principle for “Greater scrutiny of perverse industry and innovation agendas”. We do reiterate that the PPP – when one digs down into the surrounding rationales – was more broadly defined by participants, who referred less to its definition amongst expert and policy circles, and more to “polluters should pay” as a grab-bag for a range of concerns about industry and their ensuing responsibilities. Nevertheless, we attempt to take your advice in making the concepts in this paragraph more coherent – and where necessary, distinct to approach.
--	--

This manuscript covers a tremendous amount of ground, and for the most part does it quite well. I have two significant criticisms, however, one more substantive and one more stylistic.

Second, the tendency to mix statements made by focus groups together with what appear to be the authors' own inferences based on those statements (including extensions by proxy), in ways that seem to attribute the authors' understandings to focus group participants, is a persistent problem. Several examples are discussed below.

The text needs to be revised to clearly separate presentation of results in terms of participant observations from discussion of those results including their implications based on the authors' analytical framework.

Thank you for this constructive criticism, which was also raised by Reviewer 2.

This gets at the heart of a key dilemma that we faced in reporting our Results: trying to find a compromise between the scope of our data, and allowing participants to 'speak for themselves'. We are aware and concerned about the dangers of over-reduction.

In previous papers written by our group, we have adopted reporting formats that cater to participants in their own words, using quotations to illustrate or nuance, or juxtapose key positions in-text, with surrounding analytical text. We have waived the opportunity for a double-blind review, so these papers can be easily identified and sourced in open access.

□ An example would be Low, Baum & Sovacool (2022), 'Rethinking Net

Zero systems...' in *Global Environmental Change*.

□ Another would be Sovacool, Baum, Low & Fritz (2023) 'Coral reefs, cloud forests and radical climate interventions...' in *PLOS Climate*.

In this paper, we found it unfeasible to include partial or whole quotations in text, as this would produce a new set of problems: shortened quotes out of context, further narrowing from the already narrowed range of quotes available from 22 countries, and going vastly beyond the word length stipulated by Nature Communications.

A second reason is that we sought to undertake two comparisons: (1) against the existing public perceptions literature, expanding its most prevalent findings when encountering new publics, and (2) against expert assessment, innovation, and policy. (The majority of the papers in the public perceptions literature only tackles Point (1)

	in discussion, which permits more space to report results. And moreover, most public perceptions papers have between 1 and 4 countries' worth of transcripts and data.) We found that there are both gaps and overlaps between our findings and that of the existing public perceptions literature. But reporting our findings required us to foreground reference to existing public perceptions literature. Otherwise, from a linear reading, there would be an impression of plagiarism, or re-treading (some, certainly not all) old results. For both these reasons, we aimed at a degree of summary and interpretation in the Results. This is why we use the quotation tables, and themes marked by italics in text to clearly link to the same themes in the tables – where the quotations represent the fullest depiction / representation of participant observations.
--	--

	We take two actions to engage with your helpful critique. Firstly: Our intent regarding this dilemma was contained in the final paragraph of the introduction to the Results section – but we now expand and clarify further: “We compromise between two needs: summarization and reference to insights from preceding public engagement studies vs. allowing participants to speak in their own words, thereby demonstrating greater nuance, ambivalence, and discursive interaction. In text, we cater to the former need, and limit ourselves to summary descriptions of themes highlighted in italics. We recognize that this requires a degree of interpretation and reduction. However, we make use of extensive quotation tables to cater to the latter need, and to give a sense
--	---

	of the diversity and depth of deliberations. The italicized themes in text directly correspond to themes in Tables 4-6 in the supplementary materials, which also contain a count of the focus groups who spoke to them, and representative quotations. Tables 1-3, included in text, are shorter versions of Tables 4-6. These contain a selection of the most important themes and representative quotations.” Secondly: In previous drafts of this manuscript, we had experimented with adding qualifications, such as “what researchers refer to as”, or “what might be summarized by existing studies as” in reference to the key themes (or summarized concepts). In some cases, this proved repetitive and unwieldy. We have re-added these qualifications where the reviewers have suggested, and furthermore in relation to numerous terms coined by experts, such as ‘mitigation deterrence’, ‘polluter pays principle’,
	‘pollution exports’, ‘hazardous siting’, or ‘corporate social responsibility’; or by the authors, such as ‘systemic coordination’.
Figure 1—“Countries in which focus groups were represented” include only colored (non-gray) countries—is that correct? If so that needs to be stated explicitly because it’s not self-evident. It may be clearer to use a table instead.	Thank you for the suggestion. We have further clarified the figure with additional language, including: “Countries in grey were not engaged with by either survey or focus groups.”

Afforestation and reforestation— Throughout this section you write in terms of focus groups discussing, supporting, and/or opposing “ecosystems management,” but from what I can tell—including based on reading the Methods section—these groups focused specifically on A/R, not on ecosystems management more broadly, while the authors treated A/R as a proxy for ecosystems management. If I understand this correctly, then it is important to revise this section to make clear that focus groups talked about A/R and that the authors are treating this as a proxy for ecosystems management, and that focus groups did not and were not instructed to discuss the more general category ecosystems management.	Thank you for this suggestion. In an effort to streamline what might otherwise be a huge range of technical definitions, we treated afforestation and reforestation as an entry into wider and more diverse practices in managing terrestrial and marine ecosystems, communicated as ‘restoring and/or growing vegetation’. We used afforestation and reforestation as an entry term, given its inertial prominence. We take your advice, and clarify the language accordingly. ‘Proxy’ might have been poorly chosen on our part.
Line 214—This section reads as though the distinction between A/R (as CDR) and stopping deforestation (as emissions reduction) was unclear in focus groups—if so, that seems worth noting. Could participants lumping these two linked but separate approaches together have affected their deliberations?	We agree with the reviewer that avoided deforestation is technically not carbon removal. But this is a nuance (in our experience) that predominantly emerges in expert and policy conversations. In this project, we indeed found that focus groups often conflated A/R with avoided deforestation as part of general forestry management efforts. We have added some brief language to account for this. We should also note that we did not include avoided deforestation in our list of approaches or informational materials. This was brought up by participants in deliberation.
Soil carbon sequestration—Similar to the issues surrounding A/R, this section is written as though focus groups explicitly discussed biochar, when in fact they discussed SCS without specific reference to biochar while the authors treated SCS as interchangeable with biochar. If this is correct, then the text here needs to be	Thank you for this suggestion. For the purposes of streamlining approaches, we described biochar in the informational materials as part of the general SCS category, but also as something distinct. We did leave out biochar’s other applications, e.g. construction. However,

revised accordingly to avoid giving the impression that focus groups deliberated on biochar.	this means that participants did speak about biochar – though much more generally, as part of soil carbon approaches. We have added some clarifying language to the text and to the methods. However, if your recommendation would be to remove biochar, we would be happy to consider it.
Line 406—When speaking about “the underlying causes of climate change,” it is unclear whether the authors mean causes as identified by focus groups, causes as identified in the literature, and/or causes as identified by the authors themselves. This seems to continue the pattern of appearing to conflate the views of participants with the views of the researchers.	Again, thank you for raising this point. We hope that we are not conflating the views of participants with our own views. As the quotation tables demonstrate, the characterization of underlying causes of climate change and unsustainability is borne out by statements made in focus groups. As we have tried to demonstrate, this comes from the need for a degree of summarization (while linking to quote tables), and reference to previous public perceptions studies. In previous drafts of this manuscript, we had experimented with adding qualifications, such as “what researchers refer to as”, or “what might be summarized by existing studies as” in reference to the key themes (or summarized concepts). In some cases, this proved repetitive and unwieldy. We have re-added these qualifications where the reviewers have suggested, and furthermore in relation to numerous terms coined by experts, such as ‘mitigation deterrence’, ‘polluter pays principle’, ‘pollution exports’, ‘hazardous siting’, or ‘corporate social responsibility’; or by the authors, such as ‘systemic coordination’.
Line 437—Surely participants did not explicitly reference the triple helix model but rather what researchers refer to as the triple helix model?	We agree and have adopted your suggestion.

Line 608—You seem to suggest that the downside of preference for natural/naturebased CDR lies in its potential substitution for emissions cuts. But aren't its limited removal potential and nonpermanence/vulnerability to re-emission equally significant downsides?	Thank you - we agree and have added language to this effect.
---	---

REVIEWER 2

This is an admirably ambitious crosscultural study on perceptions of carbon removal. The authors are reflexive about the analytical challenges that such a research design poses and, in general, their “robust” approach is well justified. Overall, the paper has potential to make an important contribution. However, currently the positioning of study and definition of its scope and aims are quite loosely defined. This is reflected in the title of the paper, which simply describes the research design rather than summarising the study’s contribution. The Research Design section states that the aim of the analysis is to construct a global “benchmark” for deliberative research on carbon removal- this sounds like quite an important contribution and I wondered if it could be introduced earlier and perhaps elaborated? Clarifying the scope/aims would help the reader follow the analysis which, although containing many insights, sometimes seems to jump between sections and subsections without a clear connecting narrative.	We truly appreciate your insights and constructive engagement! We especially appreciate your comment that this paper contains ‘pluralist-aggregative’ reporting and analysis. Our intent is to be general and interdisciplinary, as perhaps befits Nature Communications. We report and analyse in conversation with the existing public perceptions literature, and also with wider assessment, innovation, and policy. We have undertaken your suggestion to foreground our intent to create a global, robust benchmark – this is now more prominently in the introduction, in the second paragraph. We are concerned about altering the title of the paper to reflect the global benchmark, and hope to follow the succinct, descriptive formula called for by the Nature journal constellation.
--	---

One effect of the authors' pluralist aggregative analysis, is that in presenting the Results it's not always clear when they are reporting participants' own words and when they are applying concepts (particularly in the Governance subsection).

I appreciate that this approach necessarily requires the reduction of participants' speech, but currently they only speak in their own words when they're illustrating the themes of your coding (in the boxes).

If you could show a bit more of the discursive interaction and the kinds of ambivalent speech that often emerges in focus groups, this would bolster your overarching claims about deliberative research and make the study's cultural

Thank you for this constructive criticism, which was also raised by Reviewer 1.

This gets at the heart of a key dilemma that we faced in reporting our Results: trying to find a compromise between the scope of our data, and allowing participants to 'speak for themselves'. We are aware and concerned about the dangers of over-reduction.

In previous papers written by our group, we have adopted reporting formats that cater to participants in their own words, using quotations to illustrate or nuance, or juxtapose key positions in-text, with surrounding analytical text. We have waived the opportunity for a double-blind review, so these papers can be easily identified and sourced in open access.

pluralism more tangible. It might also headoff criticisms of your aggregative approach as overly-reductive.

- An example would be Low, Baum & Sovacool (2022), 'Rethinking Net Zero systems...' in *Global Environmental Change*.
- Another would be Sovacool, Baum, Low & Fritz (2023) 'Coral reefs, cloud forests and radical climate interventions...' in *PLOS Climate*.

In this paper, we found it unfeasible to include partial or whole quotations in text, as this would produce a new set of problems: shortened quotes out of context, further narrowing from the already narrowed range of quotes available from 22 countries, and going vastly beyond the word length stipulated by Nature Communications.

A second reason is that we sought to undertake two comparisons: (1) against the existing public perceptions literature, expanding its most prevalent findings when encountering new publics, and (2) against expert assessment, innovation, and policy. (The majority of the papers in the public perceptions literature only tackles Point (1)

	in discussion, which permits more space to report results. And moreover, most public perceptions papers have between 1 and 4 countries' worth of transcripts and data.) We found that there are both gaps and overlaps between our findings and that of the existing public perceptions literature. But reporting our findings required us to foreground reference to existing public perceptions literature. Otherwise, from a linear reading, there would be an impression of plagiarism, or re-treading (some, certainly not all) old results. For both these reasons, we aimed at a degree of summary and interpretation in the Results. This is why we use the quotation tables, and themes marked by italics in text to clearly link to the same themes in the tables – where the quotations represent the fullest depiction / representation of participant observations.
--	--

	We take two actions to engage with your helpful critique. Firstly: Our intent regarding this dilemma was contained in the final paragraph of the introduction to the Results section – but we now expand and clarify further: “We compromise between two needs: summarization and reference to insights from preceding public engagement studies vs. allowing participants to speak in their own words, thereby demonstrating greater nuance, ambivalence, and discursive interaction. In text, we cater to the former need, and limit ourselves to summary descriptions of themes highlighted in italics. We recognize that this requires a degree of interpretation and reduction. However, we make use of extensive quotation tables to
--	---

	cater to the latter need, and to give a sense of the diversity and depth of deliberations. The italicized themes in text directly correspond to themes in Tables 4-6 in the supplementary materials, which also contain a count of the focus groups who spoke to them, and representative quotations. Tables 1-3, included in text, are shorter versions of Tables 4-6. These contain a selection of the most important themes and representative quotations.” Secondly: In previous drafts of this manuscript, we had experimented with adding qualifications, such as “what researchers refer to as”, or “what might be summarized by existing studies as” in reference to the key themes (or summarized concepts). In some cases, this proved repetitive and unwieldy. We have re-added these qualifications where the reviewers have suggested, and furthermore in relation to numerous terms coined by experts, such as ‘mitigation
--	---

	deterrence’, ‘polluter pays principle’, ‘pollution exports’, ‘hazardous siting’, or ‘corporate social responsibility’; or by the authors, such as ‘systemic coordination’.
Related to this, your methodology involves both a bottom-up approach to coding and an analysis informed by themes derived from perception literatures (e.g. Smith et al, Sovacool et al). You suggest these are presented separately in the Results and Discussion, respectively. But I wasn’t clear whether the “key technical and societal issues” presented in the Results section were derived from the bottom-up coding alone? Can you briefly elaborate on this, perhaps in the section on coding methodology.	Thank you for this query. We clarify in the Methods section that the key technical and societal issues presented in the Results section are derived from the bottom-up coding. We further clarify that in our Results, we briefly foreground where results agree or diverge with the public engagement literature. In the Discussion, we compare our results (a) in greater depth against the public engagement literature, as well as (b) against assessment beyond public engagement, innovation, and policy at a landscaping level.

More generally, the reason for structuring the Results in three sub-sections could be better justified and narrated. Sections 1 & 3 (the 5 technologies, governance issues) seem clearly reflected in the research design. It's less clear, at least from a linear reading, how the climate/sustainability action section was arrived at. To help the reader, this could perhaps be briefly elaborated in the Introduction.

Thank you for this suggestion.

We reference climate and sustainability action because it was (a) often brought up by focus groups, despite not being foregrounded in questioning or information materials, and (b) is a part of the existing public perceptions literature, and thus need to be compared with our results.

The Research Design and Literature Review section already contains a reference, though brief, to this effect: "Other studies point out public capacities to assess synergies or trade-offs of upscaling carbon sinks with wider climate and sustainability action (Wibeck et al., 2017; Cox et al., 2020; Raimi, 2021; Satterfield et al., 2023), including concern over the development of excuses to further delay comprehensive emissions reductions (McLaren et al., 2021); and conceptions of justice and equity that drive preferences (McLaren et al., 2016)."

We now add to this paragraph some language explicitly noting that we draw upon several topical areas from the public perceptions literature to help organize our results, one of which is CDR's relationship

to wider climate and sustainability action. We hope that this aids how this area was arrived at in a linear reading.

The Discussion section contains many interesting insights, but (perhaps because they mirror the structure of the Results) I wasn't always clear how these related to the overarching aims of the paper. The Introduction suggests that this section is where you compare your findings against perceptions literatures and expert/policy debates. But don't you also do that to some extent in the Results section? Clarifying this further would help the reader.

Thank you for pointing this out. It touches upon several dilemmas we faced in organizing the Results and Discussion.

Originally, we had COMBINED the Results and Discussions section holistically. For example, the Direct Air Capture results was immediately followed by the relevant discussion section. In internal review, we decided to follow the traditional separation of Results and Discussion in anticipation of reviewer expectations – which is why they directly mirror each other.

We had also originally had the Results simply report the data, with no reference to the CDR public perceptions literature. But in a linear reading, this made it appear as if we were plagiarizing the CDR public perceptions literature, and in our view, some notation of previous works with comparable analysis was warranted.

Much of this is difficult to qualify in the manuscript.

Hence, we clarify:

In Results, we foreground where results agree or diverge with the public engagement literature.

In the Discussion, we further compare our results to how CDR is emerging in assessment beyond public engagement, innovation, and policy, at a landscaping level.

The Conclusion presents “Four Robust Governance Principles”. These principles are well-rehearsed in the carbon removal literature, as the authors acknowledge. In this sense they are certainly robust. But the distinctive contribution of your study seems to get a little lost here. Could you perhaps say something about the aim to construct a

Thank you for this suggestion. Following your advice, we add “Following our intent to derive a global benchmark of public perceptions for informing decision-making, we note four governance principles that would be broadly robust across two axes – across the global North and South, and across the spectrum of biogenic to engineered carbon removal.”

global benchmark for deliberative research on carbon removal?	
--	--

p.4 l1146-8: rather than talking about your future plans for ‘vertical’ analysis, perhaps you could elaborate more on the contribution the ‘global, horizontal’ approach can make?	Thank you for this suggestion. We have moved the sentence on a global benchmark to a more prominent location in the introduction, where the surrounding text works as an elaboration on what a global, horizontal benchmark might be.
p.5 l194: “Focus groups in every nation”. Is nation the right term here?	We have altered this to ‘country’.
p.5 l223: “This broad range of initiatives will need to be made more practical on national and local levels.” Sounds odd to be drawing out learnings at this stage of the paper. Consider rephrasing.	We agree and have removed the sentence.
p.5 l224: “lack of technical understanding regarding the scale and calculability of urban flora” – can you be more specific e.g. do you mean in terms of carbon sequestration?	Thank you – we have altered the language to “the scale and sequestration potential of urban carbon stocks”.
p.6 l231: “strong global pluralities”. This means a plurality of groups across Global North/South right?	It does, and we have made the language reflect it.

p.7 1275: “a surprisingly common concern was on soil toxicity”. Why surprising and why can it be questioned as a technical misunderstanding?	Thank you for pointing this out. Reviewer 3, in another location, asked why we mention certain technical misunderstandings. According to some experts we engaged with, soil toxicity – in the way discussed by certain participants regarding SCS, is not technically possible. Fixing carbon in soil does not poison the soil. We debated two options: First, to alter the text to: “Groups intermittently raised a concern over soil toxicity – fixing carbon in soil would alter soil composition and harm agriculture production – that is not technically possible. Nevertheless, such a concern reflects unease about the potential leakage of carbon from storage that cut across most carbon removal approaches, as well as widespread dependence on the agricultural sector.” Second, to delete the text, and a couple of other instances in which we questioned participants’ technical misunderstandings.
---	---

	For now, we have chosen the second, to account for both your comments and that of R3’s.
p.7 1277: “carbon leakage” – are you referring here to policy/accounting debates about this concept? Perhaps briefly clarify. (this question applies to several other points where you use the term)	Thank you for this – indeed, leakage has a technical and political meaning. We are referring to leakage of carbon from storage or transportation. We clarify this throughout the text.
p.7 11280-2: “participants found DACCS more difficult to technically grasp, while seeing it as a centralized, supply-driven cooperation between government and industry”. Could you be more specific e.g. was it the chemical removal process?	Thank you – we clarify that this was not about the process but about the infrastructure, energy provision and siting.

p.7 p.302: “NIMBYism unsurprisingly emerged in global North groups”. I’m clear whether you are characterising some of your participants as NIMBYs here? This term has been much critiqued- consider using a less contentious concept?	Thank you for pointing this out. We do not intend to characterize participants or focus groups (mostly in Europe) as NIMBYs, but rather point out that participants are grappling with the concept of NIMBYism – often using wind turbines and other such antecedent debates. We alter the language accordingly. We are aware of the contentiousness of the NIMBY term, although we note this in the discussion: “We should be wary of dismissing leakage concerns as NIMBYism. Wind turbines can be a red herring – social acceptance and opposition is tied up not only by proximity, but by the kind of infrastructure or system component, locality and vulnerability, and trust in governing institutions (e.g. Carley et al., 2020; Satterfield et al. 2023; Scott-Buechler et al. 2023).”
p.8 ll334-6. As above, when you say that “technicalities” were hard to grasp it would help if you were more specific.	Agreed, we have clarified that we mean technicalities of “carbon drawdown as well as storage”, which are the words following.
p.9 ll389-90. Technical misconception about “double removal”. Hasn’t double counting in BECCS supply chains also been raised as a concern by some experts?	It has! But this isn’t a misconception about carbon drawdown processes, which is the subject of this paragraph.
p.12 l436: “Participants ubiquitously referenced a high degree of trust in expertdriven assessment, often referring to the ‘triple helix’ model”. I’m curious about how this was expressed, particularly in countries where science plays a less prominent role in public life.	Thank you for this suggestion. In broad strokes, this was described similarly regardless of country or North vs. South. The example quote given is from the Urban group in Indonesia, but it is similar to such quotes from global North and other groups.
	We too are curious of how difference might emerge in more in-depth analysis, but hope that we might reserve this for a later investigation. We are worried here about what the analysis of science in public life across 22 countries might do in the present manuscript structure.
p.18 l582: The title of the Discussion section comprises a set of quite broad themes. Could the focus be clarified?	Thank you. We have altered the title of the section to: “Comparing public perspectives against expert assessment, innovation, and policy.

p.21 1845: “Subsidiarity would appear to be a justifiable governance principle in operationalization...”. I didn’t follow this point.	Thank you – we have added in a brief definition of subsidiarity.
p.22: The diagram accompanying the conclusions, though complicated, is potentially a useful roadmap for the reader- would it make sense to introduce this earlier?	We appreciate this suggestion! This figure summarizes (and simplifies) the entire paper, and as such, poses dilemmas wherever it is placed. We agree with the reviewer, but if we placed Figure 2 earlier, the reader lacks a reference point in the written text for much of the content. Moreover, it would compete for space with the quotation tables. We hope that we can leave it where it is.
p.26 11099: Can you briefly justify the choice to use of Zoom for the focus groups instead of a more traditional face-to-face set up? I appreciate there are obvious practical and resource-related reasons for doing so, but there is also a lot of methodological discussion that you could signpost here.	Thank you! We clarify that this choice was for ease of logistics, cost, recording and transcription.
p.26 11101: If I understand right, most focus groups discussed carbon removal for around 1 hour. That’s not a long time at all, relative to other deliberative research on carbon removal. I appreciate you gave materials in advance and allowed participants to discuss and research the topic. Could you say something briefly to qualify this approach to deliberative research?	Thank you for this suggestion. We have added some qualifying text to the methods section. Indeed, this was a need to compromise between our financial resources, and the inclination of our research design towards greater geographic coverage, and to set a global benchmark for further deliberation.

REVIEWER 3

This is a useful and interesting study of public perceptions of carbon removal (CR). Its strength is in its global reach and breadth, with 44 focus groups held in 22 countries. Its findings do not necessarily present any surprises, as the authors acknowledge, given that public perceptions of CR were already well studied. However, the international focus – and particularly, applying the same methodology across all countries, is novel. The paper offers some sound advice on governance principles for CR which again are in line with similar previous work. I recommend it for publication if the issues addressed below are considered.	Thank you for your constructive engagement!
Clarity about the purpose of consultation. Why were people being asked for their views on CR? Many CR technologies are not directly consumer-facing, ie, unlike in other domains eg transport or home energy, they do not require changes in behaviour or actions. Therefore (as with other technologies eg grid infrastructure) the point of investigating public perceptions is not to influence consumer uptake. Is the aim to guide policy support for and governance of CR, as the conclusions imply? If so, this should be made more explicit in the initial parts of the paper.	Good point. We had intended for the third paragraph of the introduction to serve as the ‘purpose of consultation’, and have now tailored the language to make this especially clear. We follow a literature of antecedent study in climate action, technology governance, energy etc, in seeing deliberative engagement as valuable for “mapping ‘situated’ perspectives (bottom-up actor- and locale-specific; in distinction to topdown, systemic, global-planning), anticipating the ‘fit’ between emerging issues and local context, and developing societal capacity for further inquiry and learning-by-doing”, especially in contexts that “cross socio-political, technological, and ecological boundaries, and require existing and novel sectors and practices to be integrated”. Accordingly, “we deploy deliberative engagements to anticipate key challenges and provide input for shaping legitimate governance processes.”

At times in the paper, it seems as if the aims of the consultation were to compare public perceptions with stakeholder perceptions (or ‘lay’ vs ‘expert’ as commonly described). If this is the case, this is valuable – for example, publics may – and usually do – prioritise notions of fairness and justice,	Thank you for this request for clarification. Regarding the bracketed text, beginning with “(However, the paper also...)” It is not our intent to prioritize technical understanding, nor to pose correcting of misconception as part of the deficit model.
---	--

which is an important lens through which to consider governance. (However, the paper also comments on how well publics ‘understood’ or ‘grasped’ the fundamental technical aspects of different CR techniques. I am not sure of the purpose of this. Given that many CR techniques are not yet commonplace, it is hardly surprising that understanding is low. At times, the paper seems to lapse into judgement of public perceptions which are deemed to be based on inadequate knowledge, eg line 275 ‘a surprisingly common concern was soil toxicity’ / 318 ‘strangely, the pollution export concern was raised in almost no...’ / 792 ‘troublingly, there was almost no awareness...’ This poses the danger of a ‘deficit model’ of public engagement, where the focus is on ‘correcting’ public perceptions through correcting a perceived information deficit.) Is it an explicit aim to compare – and evaluate – publics’ understandings compared to stakeholders/experts? Or is the aim more a democratic one, to ensure that governance is informed by public views and values?	We do not intend to place technical understanding above social understanding. In most cases, both technical and social understandings reflect intertwined, evolving, and incomplete conversations. Our intent with regard to the examples given in the bracketed text is: If there are especially key, basic misunderstandings of what an approach does, then this conditions how we should engage with hopes and concerns that follow from such deliberations. Regardless, after some consideration, we have removed the text on soil toxicity. There is a second – and more significant – rationale behind the language parsed by the reviewer. We see it as essential – especially in aiming at an interdisciplinary and general audience – to relate a ‘global benchmark’ of publicly perceived challenges and preferences to how those same conversations are emerging in expert, innovation, and policy circles. We want to point out overlaps, but importantly, the gaps. The gaps do not represent an information deficit to be filled with more technical information, but inequities in which publics require more input into decision-making.
---	---

On the methodology, the chosen method was focus groups, yet the project did not seek to capture spontaneous views on CR, but provided upfront information on different CR techniques and approaches, as detailed in the annexe. This is actually moving toward more deliberative research (which specifically involves a 'learning phase') and considerably alters the likely outcomes. I can see the reasoning for providing basic information about CR technologies, given that there is little lived experience of them, but the fact that the study involved a 'learning phase' / informational input / stimulus materials needs to be stated upfront.

Thank you for pointing out this important nuance. We adopt your suggestion and clarify this in the methods section.

The use of the term 'manpower' is outdated – could the gender neutral term 'labour' be used instead?

Agreed, and thank you – we have made this change.

Line 228 – limited discussion of offsetting – this is very interesting, and relevant to questions of governance – could it be explored further?

This point here – connected to an additional point in the discussion – is that offsetting, credits, and markets are barely mentioned at all by any group. It is a priority that rather emerges in assessment, innovation, and policy.

But in a linear reading, this sentence is abbreviated and unedifying. We delete it in the Results section, but refer to it at greater length in the Discussion:

“Moreover, there is a latent international dimension of inequity and burden-shifting, with the greatest (modelled) capacity for biogenic sequestration in tropical forested countries (Strefler et al., 2021; IPCC, 2019) – raising questions of impacts on local communities (Dooley et al, 2022). REDD+, the financing mechanism for projects in the global South to reduce deforestation (West et al., 2020), as well as the emphasis of voluntary carbon markets on forestry credits (Greenfield, 2023), have chequered histories in emissions accounting. Indeed, regarding carbon markets: offsets were sparingly mentioned in European groups and not at all in global South groups, hinting at a profound gap between policy and public awareness. Negotiations to develop rules for international carbon credits, rigorous accounting, and potentially incorporate REDD+ are ongoing over Article 6.4 of the Paris Agreement, as well as for bilateral trading of credits in Article 6.2.”

291 – perceptions of government-industry collaborations – is this influenced by deeper questions of trust in government?

Thank you for this query. ‘Trust in government’ came across much more implicitly and opaquely, in this regard. The trust in industry and innovation was clearer, and in kinds of state-owned enterprises. But trust in government as a steering factor was less clearly spoken to (except in China), and we are reluctant to make generalizing conclusions about trust in government regarding countries as diverse as China, India, Saudi Arabia, Norway, and Switzerland.

305 and elsewhere – might be worth stressing that the term ‘leakage’ refers to actual physical leakage of stored gases (I think) – compared to the often-used phrase ‘carbon leakage’ in a metaphorical sense, ie imports/exports of embodied carbon in products.	This was raised by another reviewer as well. We have changed our discussion of leakage accordingly.
402 and 625 – concerns about CR tended to be linked to localised concerns – were the groups provided with any information about global carbon budgets or the Paris-agreed target of net zero emissions? This is likely to have a significant impact on views of CR. i.e. there is no IPCC global pathway (or other significant modelling) which forecasts achievement of the net zero target without a certain amount of CR. Knowledge of this is likely to significantly influence people’s attitudes toward CR (as seen, for example, in national Citizens’ Assemblies on climate change. Without background understanding of carbon budgets, people may see CR as one among a range of different options for meeting net zero targets.	Thank you for noting this. This information was not provided, and therefore should have played no steering role.

REVIEWER COMMENTS

Reviewer #1 (Remarks to the Author):

As I read your responses and re-read your manuscript, it is increasingly clear to me that your (understandable) desire to squeeze all your findings into a single article cannot be accommodated without sacrificing too much in the way of analytical and linguistic clarity and theoretical insight. This fundamental issue was reflected in my two previous criticisms, neither of which I regard as satisfactorily addressed.

Regarding my more substantive criticism, I'm disappointed that you were unable to sharpen your analysis by drawing a sharper distinction between biogenic and engineered approaches; while I appreciate that you've added nuance to your discussion of principles, there remains nothing systematic or categorical about your treatment. Consequently, I still think your principles are too generic—to what significant environmental problem would governance rationales of “prioritize public engagement, industry and corporate agendas, systemic coordination, and underlying and interrelated causes” not apply? I trust that you “tried to derive separate lists of principles,” but it's hard for me to accept that coming up with separate lists that differ in meaningful ways is unachievable. To take one obvious example, wouldn't a governance principle related to ensuring permanence be both primarily applicable to biogenic methods and clearly warranted?

I also appreciate your altering “polluter pays” to “industry and corporate agendas,” but unfortunately this change ends up reinforcing my point. The PPP is a specific idea, but your commitment to aggregation stretched it beyond the point of recognition. You could have addressed this in two different ways. First, you could have loosened your commitment to lumping and specified that the PPP is particularly relevant to a subset of CDR approaches; in essence that was my recommendation. Or second, you could have broadened the principle to be maximally inclusive in the context of CDR; this is what you did. But the result, “greater scrutiny of industry and corporate agendas,” is relevant to a huge number of social/political/economic/ecological problems—what use is something that is so widely applicable?

Regarding my more stylistic criticism about confusing participant and author perspectives, the obvious solution is to have separate results and discussion sections, the former presenting what respondents said in their own words with minimal editorializing, the latter interpreting these results including in relation to prior research. I appreciate that you have re-inserted language to clarify when participants did not in fact use terms or concepts that otherwise appear attributed to them, but unfortunately I don't think this goes far enough: individual sections that simultaneously seek to 1) present results, 2) characterize these results in terms of an analytical framework participants were neither familiar with nor used, and 3) engage with other researchers at a theoretical level, is simply trying to do too much and leaves the reader confused.

You note in your response that “In this paper, we found it unfeasible to include partial or whole quotations in text, as this would produce a new set of problems: shortened quotes out of context, further narrowing from the already narrowed range of quotes available from 22 countries, and going vastly beyond the word length stipulated by Nature Communications.” Perhaps the problem, then, is attempting to cover all this ground in a single peer-reviewed article. Maybe you should consider a different article type, a different journal, a series of linked articles (“Negative Emissions” parts 1, 2, and 3 published in ERL in 2018 comes to mind), or a working paper. Giving yourselves more room to work with would allow you to present and discuss results in a more methodical way that maintains analytical distinctions and conceptual clarity while being more reader-friendly. And it would allow you to focus more on the distinction between natural and technological as it applies to governance principles—after all, in your response to my first criticism you write that “Taking your recommendation requires lengthening, as well as some repetition with previous sections – but we are happy to do.”

In fact, you don’t seem happy to do this, but rather seem to double down on your approaches to principles and presentation, that is, excessive aggregation, compression, reduction, and streamlining, to fit everything into a single manuscript. I understand this urge, but in this case, I think indulging it is a mistake that results in an overstuffed and confusing paper that contributes less to the literature than an alternative version or versions could and should. I cannot recommend publication.

Minor issue—With regard to biochar, the key point is that SCS and biochar are not the same thing: biochar adds carbon while SCS promotes carbon uptake. You describe “Soil carbon sequestration (including, but not interchangeable with biochar),” but SCS simply does not include biochar. Quality control should have caught this. I do not recommend removing biochar, since it was communicated and discussed, but at a minimum this mistake should be acknowledged, perhaps in a footnote in the main text and at greater length in methods.

Reviewer #2 (Remarks to the Author):

I’m satisfied that the authors have considered the comments raised in my review. In particular, the foregrounding of the global ‘benchmark’ for deliberation in both the Introduction and Conclusion helps make clearer an important intended contribution. For greater impact with the community of perceptions and deliberative researchers, the authors might consider further elaborating this ambition e.g. how would they see the benchmark being applied in deliberative practice?

The methodology discussion at the top of the Results section now elaborates the distinctive analytical approach taken by the authors. You now note that you “recognize this requires a degree of interpretation

and reduction. However...”; since thematic reduction is essential to your analysis you could perhaps be more assertive here. Could your interpretative approach be justified in more substantive terms? (e.g. in relation to your benchmark aim, for instance)

Various references to “technical misunderstanding” have now been removed from the text, but the authors still maintain that various technicalities were hard to “grasp”. The additional qualifications added show that different things are going on when participants struggle to grasp technicalities i.e. it’s not necessarily a cognitive matter. For this reason, you might consider revising the title of the Discussion subsection: “Less understanding... of EW, BECCS and DACCS”.

As before, I’m happy for the authors to contact me directly to discuss or clarify anything above.

Reviewer #3 (Remarks to the Author):

Thank you for the amendments to the paper and for your comments on the reviews. I think the paper is of an appropriate standard for publication and so have no further suggestions.

Reviewer 1

As I read your responses and re-read your manuscript, it is increasingly clear to me that your (understandable) desire to squeeze all your findings into a single article cannot be accommodated without sacrificing too much in the way of analytical and linguistic clarity and theoretical insight. This fundamental issue was reflected in my two previous criticisms, neither of which I regard as satisfactorily addressed.

Thank you for your considered review. We believe that there are some key disagreements and misunderstandings – we hope to explain the latter clearly, and perhaps bridge the former.

We ask if your suggestions are more appropriate to expert analyses of policy and governance, *rather than extrapolating principles from global public deliberations on a broad range of challenges and preferences.*

Moreover, we ask if your suggestions are appropriate for *broad governance principles in a concluding section, when clear gaps and overlaps are established in previous sections of results and analysis.*

As we had already written: “Publics cannot be expected to speak of policy mechanisms or governance institutions with the same detail as experts or decision-makers. Rather, groups pinpoint rationales for guiding policy, or archetypes of local, national, or international mechanisms and institutions.”

Regarding my more substantive criticism, I'm disappointed that you were unable to sharpen your analysis by drawing a sharper distinction between biogenic and engineered approaches; while I appreciate that you've added nuance to your discussion of principles, there remains nothing systematic or categorical about your treatment. Consequently, I still think your principles are too generic—to what significant environmental problem would governance rationales of “prioritize public engagement, industry and corporate agendas, systemic coordination, and underlying and interrelated causes” not apply?

You ask that we draw a sharper distinction between biogenic and engineered approaches in the governance principles of our concluding section.

We appreciate your suggestion, and have accommodated it to the extent permitted by the data and study design. Please allow us to elaborate.

- a) *Please note that we did not reject your suggestion, but aimed at a compromise.* To accommodate your suggestion, we doubled the length of our conclusion, with content nuanced not only by biogenic vs. engineered, but by individual approaches, or even components and sub-types of approaches.
- b) *Deriving separate lists of principles for biogenic and engineered would not match our results.* Our results (summarized in Figure 2) shows that publics note challenges and governance preferences that *overlap* multiple kinds of approaches, although with variations. Indeed, since participants were discussing multiple approaches in parallel or in sequence, it is perhaps inevitable that they would seek comparison and find overlaps.
- c) *Your suggestion for a 'systematic and categorical' treatment of overlaps and gaps between approaches is already in the results and discussion.* The conclusion aims for synthesis, and to move away from repetition. Despite this, we have attempted to accommodate your suggestion in the conclusion.
- d) *The very point of governance principles is to be broad. Policies, rather, should be specific.* Again, we have added significant detail to the concluding principles, much of which regards policy.
- e) *Indeed, the fact that these principles reflect endemic issues makes them meaningful.* We disagree that these

principles should be discarded as 'too generic'. If such principles logically apply to "any significant environmental problem", then it should be significant that they consistently fail to be enacted, and are so consistently surfacing in aspirational global governance as to be labeled as 'generic'. There is ample potential for carbon removal to become treated as a technocratic carbon management strategy. Nothing about these principles is obvious or inevitable. We attempt to show that the hopes, concerns, and preferences that underpin these principles are globally robust.

f) *If disaggregation is your aim, you will be aware of arguments in CDR assessment and policy that a division between biogenic and engineered is not enough, and indeed, is somewhat artificial.* In this view, all CDR systems have hybrid elements, and governance should pay attention to situated, locale-relevant combinations of components rather than broad categories. Yet, you do not appear to favour principles broken down by individual approaches, or by components. (By this logic, we should derive at least 5 lists!) *There is clearly value to aggregation – we ask you to consider the value of broad principles for a concluding section.*

I trust that you “tried to derive separate lists of principles,” but it’s hard for me to accept that coming up with separate lists that differ in meaningful ways is unachievable.

To take one obvious example, wouldn’t a governance principle related to ensuring permanence be both primarily applicable to biogenic methods and clearly warranted?

We appreciate that efforts behind the scenes have to be taken at face value.

However, both of the examples (permanence and PPP) you cite give us the opportunity to illustrate the value of *broad principles* that follow from *public* (rather than expert) deliberations.

Put another way: you cite these as examples of why principles should be split between biogenic and engineered. We show – using our results – that they are examples of why

	they should be nuanced between biogenic and engineered, but not completely split. Put another way, this bundling was not driven by us, but in terms of how the publics perceive these activities. We begin with your example on permanence, and follow with your example on the PPP. (Im)permanence certainly applies more to biogenic approaches as a shorter-term, multi decadal (at best) issue, if you take it as a technical matter of (biological, chemical, etc) sequestration processes and storage periods. But if you look at how publics discuss storage duration, it has less to do with these technical processes, and more to do with (profit seeking) motives damaging the safety and reliability – and therefore, the duration - of storage. For example, for biogenic approaches, this might regard deforestation pressures, and for storage from DACCS or BECCS; it might have more to do with temporary or unsafe storage sites. Therefore: if we were to form a governance principle for this, it would certainly have to maintain distinctions between deforestation and carbon storage leakage as distinct kinds of impermanence. Nevertheless, the overarching insight and principle would need to center on public concern for storage safety and duration to be shortcut by a range of (profit seeking) motives.
I also appreciate your altering “polluter pays” to “industry and corporate agendas,” but unfortunately this change ends up reinforcing my point. The PPP is a specific idea, but your commitment to aggregation stretched it beyond the point of recognition. You could have addressed this in two different ways. First, you could have loosened your commitment to lumping and specified that the PPP is particularly relevant	This example offers a second opportunity to illustrate the value of broad principles that follow from public (rather than expert) deliberations. Again, the aggregation was not driven by us, but in terms of how the publics perceive these activities. You recommend that the PPP be specified as “particularly relevant to a subset of CDR approaches,”, by which you (likely) mean

to a subset of CDR approaches; in essence that was my recommendation. Or

high cost, infrastructure-intensive options like DACC.

second, you could have broadened the principle to be maximally inclusive in the context of CDR; this is what you did. But the result, “greater scrutiny of industry and corporate agendas,” is relevant to a huge number of social/political/economic/ecological problems—what use is something that is so widely applicable?

But publics also see a role for powerful and/or polluting companies and industries (sometimes, state-owned enterprises) to marshal their resources in biogenic approaches – for example, tree planting campaigns. PPP does not only apply to high-cost options. Rather, it is a function of wider hopes and concerns about industry – about culpability, and responsibility to lead action.

Indeed, our results reveal complexes of hopes, concerns, and preferences that may perhaps not be as clearly delineated as the contents of a policy brief might. We reiterate that the PPP – when one digs down into the surrounding rationales – was more broadly defined by participants, who referred less to its definition amongst expert and policy circles, and more to “polluters should pay” as a grab-bag for a range of concerns about industry and their ensuing responsibilities.

These differences and nuances in understanding should be further engaged with.

However, upon reflection, we agree that “greater scrutiny of industry and corporate agendas” is vague, and alter it to: *Scrutiny and regulation of the role of industry in carbon removal should be developed beyond incentivizing innovation.*

Regarding my more stylistic criticism about confusing participant and author perspectives, the obvious solution is to have separate results and discussion sections, the former presenting what respondents said in their own words with minimal editorializing, the latter interpreting these results including in relation to prior research. I appreciate that you have reinserted language to clarify when participants did not in fact use terms or concepts that otherwise appear attributed to them, but unfortunately I don't think this goes far enough: individual sections that simultaneously seek to 1) present results, 2)

We are happy to re-engage with your suggestion that we separate the Results and Discussion more clearly.

You will see that we have now separated these sections. All references to expert literature interpretations of data – mostly, the insights from previous CDR public perceptions studies – have been moved to discussion.

We remain concerned that erasing references to the voluminous public perceptions literature in the results makes it appear as if we are showing entirely novel

characterize these results in terms of an analytical framework participants were neither familiar with nor used, and 3) engage with other researchers at a theoretical level, is simply trying to do too much and leaves the reader confused.

results. Noting this literature foregrounds overlaps and gaps. Nevertheless, we have undertaken revisions so that the presentation is now cleaner.

You note in your response that “In this paper, we found it unfeasible to include partial or whole quotations in text, as this would produce a new set of problems: shortened quotes out of context, further narrowing from the already narrowed range of quotes available from 22 countries, and going vastly beyond the word length stipulated by Nature Communications.”

Perhaps the problem, then, is attempting to cover all this ground in a single peerreviewed article. Maybe you should consider a different article type, a different journal, a series of linked articles (“Negative Emissions” parts 1, 2, and 3 published in ERL in 2018 comes to mind), or a working paper. Giving yourselves more room to work with would allow you to present and discuss results in a more methodical way that maintains analytical distinctions and conceptual clarity while being more reader-friendly. And it would allow you to focus more on the distinction between natural and technological as it applies to governance principles—after all, in your response to my first criticism you write that “Taking your recommendation requires lengthening, as well as some repetition with previous sections – but we are happy to do.”

In fact, you don’t seem happy to do this, but rather seem to double down on your approaches to principles and presentation, that is, excessive aggregation, compression, reduction, and streamlining, to fit everything into a single manuscript. I understand this urge, but in this case, I think indulging it is a mistake that results in an overstuffed and confusing paper that contributes less to the literature than an

We appreciate that this paper has an ambitious scope, indeed is grounded in a uniquely wide and detailed set of focusgroup data.

Our intent here is to *provide a ‘global benchmark’ across technologies and national publics*. R2 and R3 have agreed to this as the framing logic, and R2 suggested that we highlight it especially.

This current ‘global benchmark’ paper is already greatly narrowed, and is part of a sequence of papers in preparation and in review. We do not have the space to describe all the details here – but 5 macroapproaches and 44 focus groups across 22 countries in all UN regions creates a daunting number of dimensions along which to divide the data.

If our publications were to aim at aggregation, they might be critiqued as vague and “overstuffed”; if they aim at disaggregation, they might be critiqued as salami-slicing and lacking a comparative dimension. Every reviewer or reader will have a subjective judgement on the level of (dis)aggregation – indeed, there is no shortage of internal discussions on these points.

Part of the issue is therefore the dilemma driven by the project’s scope and richness of data – in other words, one of the major contributions of the paper, given the lack of engagement with the global South and overall deficit of cross-country studies, is (to our mind, incorrectly) presented as a shortcoming.

In this paper, we fill a need for a broad ‘overhead’ paper, somewhat akin to how

alternative version or versions could and should. I cannot recommend publication.	IPCC Assessment Reports are accompanied by aggregative syntheses reports, further accompanied by much more detailed Working Group and chapter-based investigations.
Minor issue—With regard to biochar, the key point is that SCS and biochar are not the same thing: biochar adds carbon while SCS promotes carbon uptake. You describe “Soil carbon sequestration (including, but not interchangeable with biochar),” but SCS simply does not include biochar. Quality control should have caught this. I do not recommend removing biochar, since it was communicated and discussed, but at a minimum this mistake should be acknowledged, perhaps in a footnote in the main text and at greater length in methods.	Upon further inspection, we fully agree that the way this is presented in the text (note: not however to participants) could be quite misleading. We have therefore revised to: “Soil carbon sequestration (potentially in combination with biochar”. We of course appreciate and agree that these approaches are not the same thing. That being said, both represent means of storing carbon in soils and, consequently, are deployed together in many ongoing trials (see Low et al. 2022). While noting how they differ, this is why they are presented together to participants (please see Methods discussion), along with why we mentioned “biochar” here, even though this sub-section focuses on soil carbon sequestration. Reference: Low, S., Baum, C. M., & Sovacool, B. K. (2022). Taking it outside: Exploring social opposition to 21 early-stage experiments in radical climate interventions. Energy Research & Social Science, 90, 102594.

Reviewer 2

I'm satisfied that the authors have considered the comments raised in my review. In particular, the foregrounding of the global 'benchmark' for deliberation in both the Introduction and Conclusion helps makes clearer an important intended contribution. For greater impact with the community of perceptions and deliberative researchers, the authors might consider further elaborating this ambition e.g. how	Thank you for your insightful comments, particularly on the theory and practice of deliberation and sense-making in public engagements. In the conclusion, we have added some signposting about where our and other efforts might lead: "We hope that future deliberative engagements will elaborate on public perceptions, preferences, and ensuing
would they see the benchmark being applied in deliberative practice?	governance as they apply to more situated or locale-specific contexts: for example, regional portfolios of carbon removal, and demographics particularly in the global South."
The methodology discussion at the top of the Results section now elaborates the distinctive analytical approach taken by the authors. You now note that you "recognize this requires a degree of interpretation and reduction. However..."; since thematic reduction is essential to your analysis you could perhaps be more assertive here. Could your interpretative approach be justified in more substantive terms? (e.g. in relation to your benchmark aim, for instance)	Thank you for this suggestion. We have added some text to this effect.
Various references to "technical misunderstanding" have now been removed from the text, but the authors still maintain that various technicalities were hard to "grasp". The additional qualifications added show that different things are going on when participants struggle to grasp technicalities i.e. it's not necessarily a cognitive matter. For this reason, you might consider revising the title of the Discussion sub-section: "Less understanding... of EW, BECCS and DACCS".	We appreciate this suggestion, and have removed the reference to 'less understanding'.

Reviewer 3

Thank you for the amendments to the paper and for your comments on the reviews. I think the paper is of an appropriate standard for publication and so have no further suggestions.	Thank you once again for your constructive and rigorous comments.
--	--

REVIEWER COMMENTS

Reviewer #1 (Remarks to the Author):

After reading through your rebuttal, I believe I see the root of the disagreement between us: we each mean something different by “governance principles.”

The way I’m thinking about governance principles aligns closely with the approach laid out in “The ABC of Governance Principles for Carbon Dioxide Removal Policy” by Honegger et al. (2022) <https://www.frontiersin.org/articles/10.3389/fclim.2022.884163/full>, where the authors define the term governance principle as “norms that are at the beginning of the development and justification of an ensemble of policy instruments and their evaluation.” The authors put forward a long list of potential principles including, for example, “CDR policies should not weaken other mitigation efforts”; “Policies should ensure consistent accounting for CDR results applying conservative baselines and including leakage”; “CDR policies should fulfill principles of inter- and intragenerational equity (e.g., Polluter Pays or Ability to Pay)”; “Efforts should internationally be differentiated per common-but-differentiated responsibilities”. These principles are derived from experts. In line with this, I have been asking for – and expecting – greater specificity, based on the assumption that your scope included providing governance principles thus understood. This understanding is what led me to characterize your “principles” as “too generic”—I suggested drawing a sharper distinction between biogenic and engineered approaches as one way to add more substance.

You have not defined what you mean by governance principles. This is not an obvious oversight—I personally wouldn’t normally think to define such a widely used term—but it matters when I suspect what you mean is something much closer to what Honegger et al. refer to as “societal expectations”: “Societal expectations usually represent norms in a regulative sense that are not (yet) institutionalized or established as a rule. These normative expectations that something should (not) be done are expressed, for example, in stakeholder or civil society surveys and are an important component of democratic policy development” (compare to “global public deliberations on a broad range of challenges and preferences” or “how the public perceives these activities”). Compared to principles, societal expectations are more general (“broader”) and may be less fully articulated. If my suspicion is correct, then your resistance to my calls for greater specificity and precision is understandable—your scope is less than what my understanding of governance principles would suggest. In essence, my suggestion to disaggregate was an attempt to push you toward refining societal expectations into governance principles, but that was never your intent (despite—confusingly, from my point of view—your use of the term governance principles). It would also explain your emergent notion of “expert” vs. “public” governance principles (“broad principles that follow from public (rather than expert) deliberations”), the latter equivalent to societal expectations but necessitating an additional adjective when forced to share a term often associated with experts and specialists.

In short, we don't disagree that "The very point of governance principles is to be broad. Policies, rather, should be specific," but governance principles as I understand them are more focused—and more structured—than governance principles as you understand them. From my perspective, "hopes, concerns, and preferences" underpin expectations, which in turn underpin principles—your concluding section does not contain "broad principles" as for example outlined in Honegger et al. but rather something like "normative expectations," i.e., prioritize public engagement, industry and corporate agendas, systemic coordination, and underlying and interrelated causes.

Similarly, we don't disagree that your "results reveal complexes of hopes, concerns, and preferences that may perhaps not be as clearly delineated as the contents of a policy brief might," but such a clear delineation is precisely what your use of the term governance principles leads me to expect. Looking back over my own reviews, my initial, primary criticism of your manuscript was that "the paper's conclusions feel underwhelming, especially given all the work that precedes them. The four governance principles you identify—prioritize public engagement, polluter pays, systemic coordination, prioritize root causes—are certainly robust, but provide relatively little insight into how particular types of CDR should be governed." This criticism—and your failure to adequately address it (in my opinion) over subsequent rounds of review—is grounded in our differing conceptions of governance principles and resultant partially talking past one another.

The issue is ultimately one of (perceived) scope: your approach to governance principles leads me to assume a bigger scope—one that would (and here I think you would agree) exceed the boundaries of an "overhead paper"—than the one to which you aspire.

I see a couple of ways to resolve this. First, you can define explicitly what you mean by "governance principles"; I expect your definition would approximate the definition of "societal expectations" quoted above. If you take this path, you will need to note that your broader, more public conceptualization of governance principles conflicts with some others' more specific, often expert-driven conceptualizations. Ideally you would also justify why you choose to define governance principles in this way.

Second, you can replace the term governance principles with the term societal expectations, which I think is precisely what you have documented. This would be the cleanest solution and is the one I would prefer.

If you do one of these two things, I can support publication.

Reviewer #2 (Remarks to the Author):

I'm satisfied the authors have considered the criticisms in my previous review

Reviewer 1

Regarding:

After reading through your rebuttal, I believe I see the root of the disagreement between us: we each mean something different by “governance principles.”

The way I’m thinking about governance principles aligns closely with the approach laid out in “The ABC of Governance Principles for Carbon Dioxide Removal Policy” by Honegger et al. (2022) <https://www.frontiersin.org/articles/10.3389/fclim.2022.884163/full>, where the authors define the term governance principle as “norms that are at the beginning of the development and justification of an ensemble of policy instruments and their evaluation.” The authors put forward a long list of potential principles including, for example, “CDR policies should not weaken other mitigation efforts”; “Policies should ensure consistent accounting for CDR results applying conservative baselines and including leakage”; “CDR policies should fulfill principles of inter- and intragenerational equity (e.g., Polluter Pays or Ability to Pay)”; “Efforts should internationally be differentiated per common-but-differentiated responsibilities”. These principles are derived from experts. In line with this, I have been asking for – and expecting – greater specificity, based on the assumption that your scope included providing governance principles thus understood. This understanding is what led me to characterize your “principles” as “too generic”—I suggested drawing a sharper distinction between biogenic and engineered approaches as one way to add more substance.

You have not defined what you mean by governance principles. This is not an obvious oversight—I personally wouldn’t normally think to define such a widely used term—but it matters when I suspect what you mean is something much closer to what Honegger et al. refer to as “societal expectations”: “Societal expectations usually represent norms in a regulative sense that are not (yet) institutionalized or established as a rule. These normative expectations that something should (not) be done are expressed, for example, in stakeholder or civil society surveys and are an important component of democratic policy development” (compare to “global public deliberations on a broad range of challenges and preferences” or “how the publics perceive these activities”). Compared to principles, societal expectations are more general (“broader”) and may be less fully articulated. If my suspicion is correct, then your resistance to my calls for greater specificity and precision is understandable—your scope is less than what my understanding of governance principles would suggest. In essence, my suggestion to disaggregate was an attempt to push you toward refining societal expectations into governance principles, but that was never your intent (despite—confusingly, from my point of view—your use of the term governance principles). It would also explain your emergent notion of “expert” vs. “public” governance principles (“broad principles that follow from public (rather than expert) deliberations”), the latter equivalent to societal expectations but necessitating an additional adjective when forced to share a term often associated with experts and specialists.

In short, we don’t disagree that “The very point of governance principles is to be broad. Policies, rather, should be specific,” but governance principles as I understand them are more focused—and more structured—than governance principles as you understand them. From

my perspective, “hopes, concerns, and preferences” underpin expectations, which in turn underpin principles—your concluding section does not contain “broad principles” as for example outlined in Honegger et al. but rather something like “normative expectations,” i.e., prioritize public engagement, industry and corporate agendas, systemic coordination, and underlying and interrelated causes.

Similarly, we don’t disagree that your “results reveal complexes of hopes, concerns, and preferences that may perhaps not be as clearly delineated as the contents of a policy brief might,” but such a clear delineation is precisely what your use of the term governance principles leads me to expect. Looking back over my own reviews, my initial, primary criticism of your manuscript was that “the paper’s conclusions feel underwhelming, especially given all the work that precedes them. The four governance principles you identify—prioritize public engagement, polluter pays, systemic coordination, prioritize root causes—are certainly robust, but provide relatively little insight into how particular types of CDR should be governed.” This criticism—and your failure to adequately address it (in my opinion) over subsequent rounds of review—is grounded in our differing conceptions of governance principles and resultant partially talking past one another.

The issue is ultimately one of (perceived) scope: your approach to governance principles leads me to assume a bigger scope—one that would (and here I think you would agree) exceed the boundaries of an “overhead paper”—than the one to which you aspire.

I see a couple of ways to resolve this. First, you can define explicitly what you mean by “governance principles”; I expect your definition would approximate the definition of “societal expectations” quoted above. If you take this path, you will need to note that your broader, more public conceptualization of governance principles conflicts with some others’ more specific, often expert-driven conceptualizations. Ideally you would also justify why you choose to define governance principles in this way.

Second, you can replace the term governance principles with the term societal expectations, which I think is precisely what you have documented. This would be the cleanest solution and is the one I would prefer.

If you do one of these two things, I can support publication.

Response:

We thank the reviewer for the considered response, which gets at the crux of what we have – separately – been meaning with the use of “principles”. We are happy to adopt your suggestion to use “societal expectations”, that might further inform assessment and governance.

We further thank the reviewer for their rigour and patience through these multiple review rounds.

Reviewer 2

Regarding:

I'm satisfied the authors have considered the criticisms in my previous review.

Response:

We thank the reviewer again for their efforts.

REVIEWERS' COMMENTS

Reviewer #1 (Remarks to the Author):

Thank you for changing "principles" to "societal expectations" - this may seem like a minor point, but I think it substantially clarifies what you've done and sets your paper up to have a stronger impact. I recommend publication.

Response:

We thank the reviewer for their rigorous engagement with the manuscript, which has substantially improved it.